# Development of human cGAS-specific small-molecule inhibitors for repression of dsDNA-triggered interferon expression

Lodoe Lama[1], Carolina Adura[2], Wei Xie [3], Daisuke Tomita[4], Taku Kamei[4], Vitaly Kuryavyi[3], Tasos Gogakos[1], Joshua I. Steinberg[1], Michael Miller[4], Lavoisier Ramos-Espiritu[2], Yasutomi Asano[4], Shogo Hashizume[4], Jumpei Aida[4], Toshihiro Imaeda[4], Rei Okamoto[4], Andy J. Jennings[4], Mayako Michino[4], Takanobu Kuroita[4], Andrew Stamford[4], Pu Gao[3,5], Peter Meinke[4], J. Fraser Glickman[2], Dinshaw J. Patel [3] & Thomas Tuschl[1]

Cyclic GMP-AMP synthase (cGAS) is the primary sensor for aberrant intracellular dsDNA producing the cyclic dinucleotide cGAMP, a second messenger initiating cytokine production in subsets of myeloid lineage cell types. Therefore, inhibition of the enzyme cGAS may act anti-inflammatory. Here we report the discovery of human-cGAS-specific small-molecule inhibitors by high-throughput screening and the targeted medicinal chemistry optimization for two molecular scaffolds. Lead compounds from one scaffold co-crystallize with human cGAS and occupy the ATP- and GTP-binding active site. The specificity and potency of these drug candidates is further documented in human myeloid cells including primary macrophages. These novel cGAS inhibitors with cell-based activity will serve as probes into cGAS-dependent innate immune pathways and warrant future pharmacological studies for treatment of cGAS-dependent inflammatory diseases.

[1] Laboratory for RNA Molecular Biology, The Rockefeller University, 1230 York Ave, Box 186, New York, NY 10065, USA. [2] High-Throughput and Spectroscopy Resource Center, The Rockefeller University, 1230 York Avenue, New York, NY 10065, USA. [3] Structural Biology Program, Memorial Sloan-Kettering Cancer Center, New York, NY 10065, USA. [4] Tri-Institutional Therapeutics Discovery Institute, 413 East 69th Street 16th Floor, New York, NY 10021, USA. [5] Present address: Key Laboratory of Infection and Immunity, CAS Center for Excellence in Biomacromolecules, Institute of Biophysics, Chinese Academy of Sciences, Beijing, China. These authors contributed equally: Lodoe Lama, Carolina Adura, Wei Xie. Correspondence and requests for materials should be addressed to D.T. (email: dat2024@tritdi.org) or to J.F.G. (email: fglickman@rockefeller.edu) or to D.J.P. (email: pateld@mskcc.org) or to T.T. (email: ttuschl@rockefeller.edu)

nnate immunity is considered a first line cellular stress response defending the host cell against invading pathogens and initiating signaling to the adaptive immunity system. These processes are triggered by conserved pathogen-associated molecular patterns (PAMPs) through sensing by diverse pattern recognition receptors (PRRs) and subsequent activation of cytokine and type I interferon gene expression[1]. The major antigen-presenting cells, such as monocytes, macrophages, and dendritic cells produce type I interferons and are critical for eliciting adaptive T- and B-cell immune system responses. The major PRRs detect aberrant, i.e., mislocalized, immature, or unmodified, nucleic acids on either the cell surface, the inside of lysosomal membranes, or within other cellular compartments[2,3].

Cyclic GMP–AMP synthase (cGAS; gene symbol *CGAS/MB21D1*) is the predominant sensor for aberrant double-stranded DNA (dsDNA) originating from pathogens or mislocalization or misprocessing of nuclear or mitochondrial cellular dsDNA[4–6]. Binding of dsDNA to cGAS activates the synthesis of c[G(2′,5′)pA(3′,5′)p], a diffusible cyclic dinucleotide referred to as cGAMP, which travels to and activates the endoplasmic reticulum membrane-anchored adaptor protein, stimulator of interferon genes (gene symbol *STING/TMEM173*)[5–8]. Activated STING recruits and activates TANK-binding kinase 1 (gene symbol *TBK1*)[9], which in turn phosphorylates the transcription factor family of interferon regulatory factors (IRFs)[10] inducing cytokine and type I interferon mRNA expression. Type I interferons are expressed from over ten *IFNA* genes and one *IFNB1* gene[5,10–12].

The critical role of cGAS in dsDNA sensing has been established in different pathogenic bacteria[13,14], viruses[15], and retroviruses[16]. Additionally, cGAS is essential in various other biological processes such as cellular senescence[17,18] and recognition of ruptured micronuclei in the surveillance of potential cancer cells[19,20].

While the cGAS pathway is important for host defence against invading pathogens, cellular stress and genetic factors may also cause production of aberrant cellular dsDNA, e.g., by nuclear or mitochondrial leakage[21], and thereby trigger autoinflammatory responses. Aicardi-Goutières syndrome (AGS)[22], a lupus-like severe autoinflammatory immune-mediated disorder, arises from loss-of-function mutations in TREX1, a primary DNA exonuclease responsible for degrading aberrant DNA in cytosol. Knockout of cGAS in *Trex1*-deficient mice prevented otherwise lethal autoimmune responses[23,24], supporting cGAS as driver of interferonopathies. Likewise, embryonic lethality caused by deficiency of *Dnase2*, an endonuclease responsible for degradation of excessive DNA in lysosomes during endocytosis, was completely rescued by additional knockout of *Cgas*[24] or *Sting*[25]. These observations support cGAS as a drug target and inhibition of cGAS may provide a therapeutic strategy for preventing autoinflammation and treating diseases such as systemic lupus erythematosus (SLE) with involvement of anti-dsDNA antibodies[26].

Efforts to develop inhibitors of cGAS and Sting have therefore been undertaken using either mouse (m) or human (h) cGAS proteins and structures. An in silico screen based on a m-cGAS-dsDNA co-crystal structure suggested the anti-malarial drug quinacrine as a potential cGAS inhibitor; however, the drug was not tested in mouse cells and an IC$_{50}$ of 3.7 μM was reported using human THP1 cells[27]. Subsequently, quinacrine was shown to indirectly affect the cGAS activity through disruption of dsDNA conformation rather than directly binding and inhibiting the enzyme[28]. The polyanionic drug suramin, approved for treatment of African sleeping sickness and river blindness, was also recently reported as biochemical and cell-based inhibitor of h-cGAS; however, inhibitory constants were not determined[29]. Fragment-library-based screening using h-cGAS followed by hit

optimization resulted in a biochemically active inhibitor (IC$_{50}$ = 4.9 μM) unable to inhibit dsDNA sensing in THP1 cells[30]. Finally, chemically reactive inhibitors for mouse and human STING crosslinking to the sulfhydryl group of Cys91 blocked cGAS-dependent interferon production in THP1 cells and mice[31]. Although inhibitory constants were not determined, concentrations of 0.5 μM specifically blocked cGAS/STING- but not RIG-I-mediated interferon production.

Our laboratories previously identified the small molecule inhibitor **RU.521** blocking m-cGAS (IC$_{50}$ = 0.11 μM) biochemically as well as in mouse macrophages (IC$_{50}$ = 0.70 μM) without interfering with other innate immunity pathways. We also provided crystallographic evidence of drug binding at the active site[32]. Here, we show that **RU.521**, despite its promise in mouse studies, is a poor inhibitor of recombinant h-cGAS. Although the mechanism of dsDNA sensing and cGAMP production are conserved between mouse and human, h- and m-cGAS only share 60% amino acid identity suggesting drug screens specifically targeting h-cGAS are valuable.

We therefore adapt a high-throughput screen (HTS) for the discovery of inhibitors of recombinantly produced h-cGAS using a chemiluminescence assay, which measures ATP consumption, and is faster and more cost-effective compared to our previously reported mass-spectrometry-based assay[32]. We describe medicinal chemistry approaches for derivatizing the screening hits into potent and specific inhibitors for h-cGAS arriving at compounds active in major interferon-producing cell types including primary human macrophages. We have also solved co-crystal structures of our best inhibitors with h-cGAS to further our mechanistic understanding of cGAS function in human. Our new inhibitors pave the way for the development of small-molecule therapeutics against cGAS-related autoimmune diseases.

## Results

**High-throughput screening assay development.** We have previously reported a high-throughput assay, which was used to identify and characterize small-molecule inhibitors of recombinant m-cGAS[32]. The mouse enzyme was activated by a 45-bp dsDNA (Fig. 1a) and reactants and products were quantified by Agilent RapidFire mass spectrometry (RF-MS). The reaction conditions suitable for screening of m-cGAS inhibitors, however, were suboptimal for screening of the less catalytically active h-cGAS (Supplementary Figure 1a). However, increasing the length of dsDNA activator from 45 to 100 bp, drastically improved h-cGAS activity (Supplementary Figure 1b). Under optimized conditions our previously developed m-cGAS inhibitor **RU.521** was 15-fold less potent against h-cGAS (IC$_{50}$ = 2.94 μM) prompting us to initiate screening efforts specifically targeting h-cGAS (Supplementary Figure 2).

Because the RF-MS assay throughput was limited by the MS reading time (1 data point per 13 s, 8 plates per day)[32], we adapted a faster luminescence-based detection method (LUM) originally utilized for screens targeting protein kinases (kinase-Glo)[33] monitoring consumption of ATP (5 data point per second, 24 plates per day). We determined the effective concentration (EC$_{50}$) for 100-bp dsDNA activating h-cGAS at 1.2 and 1.6 nM using RF-MS (Fig. 1b) or LUM (Fig. 1c), respectively. We repeated a pilot screen by incubating 100 nM h-cGAS, 25 nM 100-bp dsDNA, 100 μM ATP, and 100 μM GTP for 7 h at room temperature (RT) testing 1268 compounds from the Sigma Aldrich LOPAC compound collection in two consecutive days. The results compared with a linear regression coefficient of 0.89 (Fig. 2a), permitting us to proceed with screening of 281,348 compounds at 12.5 μM (Supplementary Table 1, Fig. 2b). We also screened all libraries against m-cGAS by the LUM assay

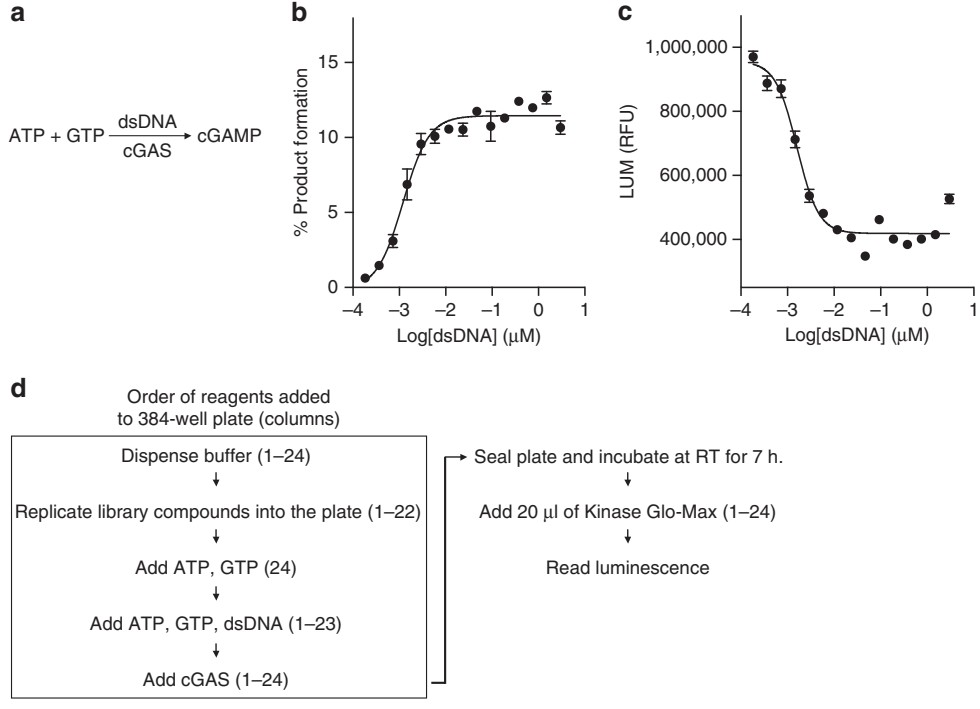

**Fig. 1** Establishment of a high-throughput screening assay for identification of small-molecule h-cGAS inhibitors. **a** Schematic showing dsDNA-activated cGAS synthesis of cyclic GMP–AMP (cGAMP) from ATP and GTP. **b, c** Assessment of enzymatic activity of h-cGAS under varying dsDNA concentrations based on RapidFire mass spectrometry (RF-MS) assay (**b**) or ATP-coupled luminescence (LUM)-based assay (**c**). **d** Schematic for high-throughput screening of small-molecule h-cGAS inhibitors using assay (**c**). Numbers in parentheses indicate the column(s) in plate into which the specified reagent was dispensed. RT room temperature. ($n = 6$ for **b** and **c**; mean ± S.D.)

using 60 nM m-cGAS and 300 nM 45-bp dsDNA for a 1-h incubation period. The screening results have been deposited into NCBI PubChem screening library database.

The h-cGAS screen identified 218 compounds with a normalized percentage of inhibition (NPI) ≥ 40. The dose response of these compounds was determined from using ten consecutive half-dilution points ranging from 25 μM to 50 nM. A total of 119 compounds were identified with $IC_{50} \leq 10$ μM, of which 19 were eliminated through comparison to the PAINS filter[34]. The remaining 100 compounds were subjected to a dsDNA intercalation test[32], which excluded another 25 compounds (Supplementary Data 1). The remaining 75 compounds were analyzed by HPLC-MS to determine their purity. Forty-six compounds showed the correct molecular mass with a purity ≥ 85%.

We repurchased 27 of the 46 compounds according to commercial availability and structural diversity and determined the $IC_{50}$ values for these compounds against h- and m-cGAS by the RF-MS method (Supplementary Table 2). We then selected the two most cross-species-active compounds (RU-0179423_AA-001, hereafter **J001** and RU-0273458_AA-001, hereafter **G001**, Fig. 3) with $IC_{50}$ values ranging from 0.5 μM to 3.8 μM for further characterization and optimization of drug-like properties.

**Characterization of J chemotype compounds**. The hit compound **J001** (Fig. 3a) is characterized by a nitropyrazole and a chloropyridine ring linked to an amide core. The $IC_{50}$ values were determined as 1.04 μM and 0.490 μM for h- and m-GAS, respectively. Nitro-aromatic ring compounds, however, are known to undergo reductive biotransformation into primary amines involving carcinogenic and genotoxic intermediates[35]. We therefore examined the possibility of replacing the nitro group by different substituents; however, none of the replacements were tolerated (Supplementary Figure 3a). Substitution of the aromatic

nitrogen atom by aromatic carbon in the pyridine moiety reduced inhibitory activity (Supplementary Figure 3b, **J005**), and the replacement of aromatic carbons by additional aromatic nitrogen atoms was either neutral (**J037, J036**) or marginally reduced the activity (**J046** or **J043**). Substitution of chlorine by hydrogen in the pyridine ring was neutral (Supplementary Figure 3b, **J038**).

We then assessed the influence of substituents at the 4 available carbon positions of the pyridine ring by employing high-throughput organic synthesis (HTOS) (Supplementary Figure 4). Substitutions at position $X_4$ abolished the inhibitory activity whereas substitution at position $X_3$, with the exception of methyl, improved the inhibitory activity. The phenyl-substituted pyridine analog **J014** yielded an $IC_{50}$ of 100 nM and 60.0 nM for h- and m-cGAS, respectively (Fig. 3a–c). The derivative **J014** then underwent HTOS substituting the nitropyrazole ring. Only the replacement by a triazole ring was tolerated (**J073**) weakening the $IC_{50}$ to 4.30 μM for h- as well as m-cGAS (Fig. 3a–c). In summary, chemical derivatization was able to improve the activity 10-fold, while the liability of a nitro group could not be corrected.

**Characterization of G chemotype compounds**. The hit compound **G001** (Fig. 3d) is characterized by a pyridoindole tricyclic core with a dichloro-substitution on the phenyl portion of the indole moiety and a methoxy-ethanone side chain connected to the nitrogen atom of the piperidine ring. The $IC_{50}$ values were determined as 2.08 μM and 0.440 μM for h- and m-cGAS, respectively. Removal of the carbonyl oxygen and methoxy oxygen or their rearrangement at different positions abolished inhibitory function (Supplementary Figure 5a, **G002, G004, G007, G008**). Likewise, introduction of bulky or aromatic groups such as phenyl (**G010**) or pyridine (**G014**) rings at the end of the methoxy-ethanone side chain eliminated inhibition. Substitution

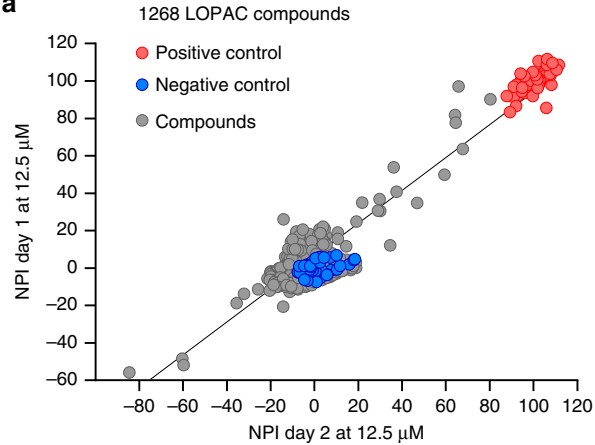

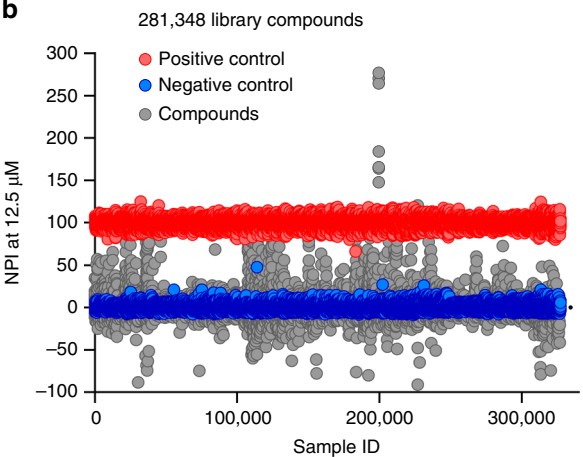

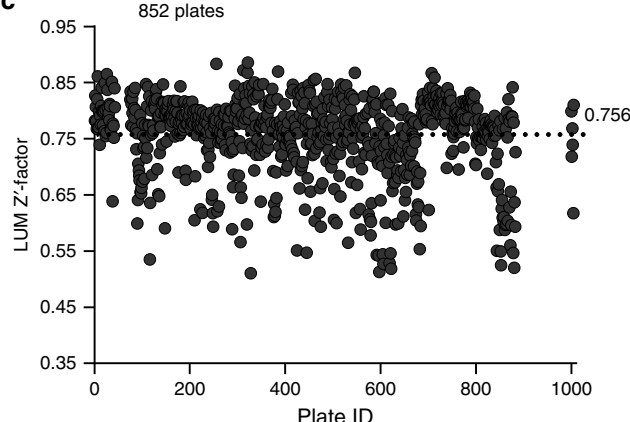

**Fig. 2** ATP-coupled luminescence-based high-throughput screening results for h-cGAS. **a** Sigma-Aldrich LOPAC (library of 1268 pharmacologically active compounds) were tested at 12.5 μM final concentration for h-cGAS LUM assay. Normalized percent of inhibition (NPI) for each compound tested (gray dots) in two independent days were plotted against each other. Blue dots indicate DMSO negative control and red dots indicate the positive control (no dsDNA). **b** Scatter plot for the NPI of the 281,348 library compounds tested at 12.5 μM final compound concentration. Sample ID indicates the unique number assigned for each compound in the library. Gray dots, library compound; blue dots, negative control; red dots, positive control. **c** Z-prime factor analysis of each library plate. Plate ID indicates the unique number assigned for each library compound plate

of the indole N–H, a potential hydrogen bond donor, by *N*-methyl was well tolerated (**G003**) but placement of bulkier group such as trifluoroethyl (**G009**) weakened inhibition.

We next assessed the influence of substituents at the four available carbon positions of the phenyl ring of the indole moiety of **G001** (Supplementary Figure 5b). Shuffling the chlorine from position $X_3$ in parental compound **G001** to position $X_2$ resulting in **G015** improved the $IC_{50}$ values 6- and 16-fold to 350 nM and 26 nM for h- and m-cGAS, respectively. Substituting chlorine with other halogens resulted in similar inhibitory activity (**G025**, **G028**), whereas bulkier substituents such as cyano (**G076**, **G078**) and methoxy (**G024**, **G026**) diminished or abolished inhibition, respectively.

We assayed the stability of compound **G015** in mouse and human microsomes and observed extensive metabolization. In mouse microsomes, we detected *O*-demethylation at the methoxy-ethanone side chain yielding 1-(6,7-dichloro-1,3,4,5-tetrahydro-2*H*-pyrido[4,3-*b*]indol-2-yl)-2-hydroxyethan-1-one (Supplementary Figure 6, **G022**) as major metabolite. While the $IC_{50}$ value of **G022** and **G015** was similar for m-cGAS, it improved 3-fold to 106 nM for h-cGAS (Supplementary Figure 6). We re-evaluated the effect of substitutions originally placed into **G001** for the metabolite **G022** and no further improvements were noted (Supplementary Figure 6).

Substitutions at the $X_4$ position of the indole phenyl ring of **G022** differentially impacted the inhibition of m- and h-cGAS. The placement of a methoxy group (Supplementary Figure 6, **G098**) resulted in a 2-fold improvement for inhibition of h-cGAS but not m-cGAS, while the placement of a pyrazole group (**G108**) yielded a 4-fold improved $IC_{50}$ value for h-cGAS to 27.5 nM but a 300-fold reduction to 5.15 μM for m-cGAS (Fig. 3d–f). Substituting the pyrazole group with methyl pyrazole (**G140**) further improved the potency yielding $IC_{50}$ values of 14.0 nM and 442 nM for h-cGAS and m-cGAS, respectively. Substituting the pyrazole group with a 2-amino pyridine ring (**G150**) resulted in an $IC_{50}$ value of 10.2 nM for h-cGAS but a complete loss of inhibition of m-cGAS. For comparison, our best inhibitors of m-cGAS are **G022**, **G086**, **G092**, **G097**, and **G098**, with $IC_{50}$ values below 24 nM, and corresponding h-cGAS values above 39 nM. The **G** scaffold inhibitors of m-cGAS are also more potent than the previously developed **RU.521** with a reported biochemical $IC_{50}$ of 110 nM[32].

**Crystal structure of the inhibitor G108 bound to apo h-cGAS**$^{CD}$. In contrast to m-cGAS, which requires dsDNA binding to drive the conformational change for opening the ATP and GTP binding pocket, h-cGAS crystallized in the open conformation in absence of dsDNA[30,36]. The h-cGAS apo catalytic domain (designated h-cGAS$^{CD}$) has therefore been used for screening of small-molecule inhibitors in absence of dsDNA[30,36]. To explore the structural basis for h-cGAS inhibition, we obtained crystals of **G108** (Fig. 4a) bound to h-cGAS$^{CD}$. During optimization of crystallization conditions, we realized that double substitution of amino acids K427E/K428E facilitated crystal packing and improved crystal quality and inhibitor occupancy substantially compared to wild-type apo h-cGAS$^{CD}$. The structure of the **G108**-h-cGAS$^{CD}$(K427E/K428E) complex at 2.45 Å resolution was solved by molecular replacement based on a previous structure of apo h-cGAS$^{CD}$ (PDB: 4O68) and refined to $R_{free}$ of 25.6% and $R_{work}$ of 22.2% with good stereochemistry (X-ray statistics in Supplementary Table 3). The overall structure of apo h-cGAS$^{CD}$(K427E/K428E) with bound **G108** in a space-filling representation is shown in Fig. 4b, with the bound **G108** occupying a similar location as bound ATP in the catalytic pocket[7,36]. The electron density in the 2Fo–Fc map (contoured at

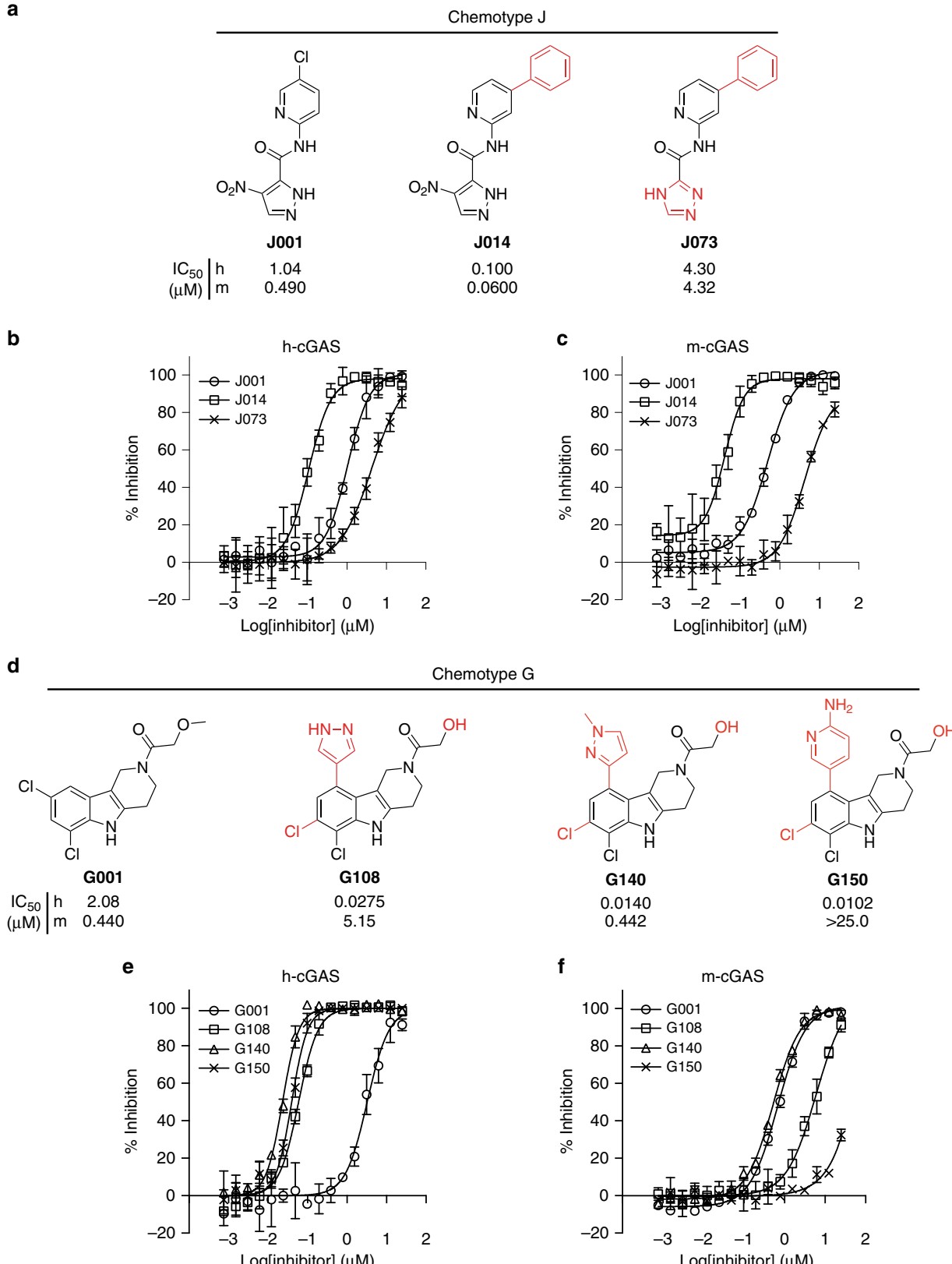

**Fig. 3** cGAS inhibition by hit compounds **J001** and **G001** and their derivatives. **a**–**c** Chemical structures indicating IC$_{50}$ values against human (h) and mouse (m) cGAS (**a**) and in vitro concentration response curves (**b** and **c**) for compound **J001** and its most active analogs. **d**–**f** Chemical structures showing IC$_{50}$ values against cGAS (**d**) and in vitro concentration response curves (**e** and **f**) for compound **G001** and its most active analogs. IC$_{50}$ values were determined for the in-house synthesized compounds using RF-MS-based assay ($n = 3$ for **b**, **c**, **e**, and **f**; mean ± S.D. Data shown are representation of two independent experiments.)

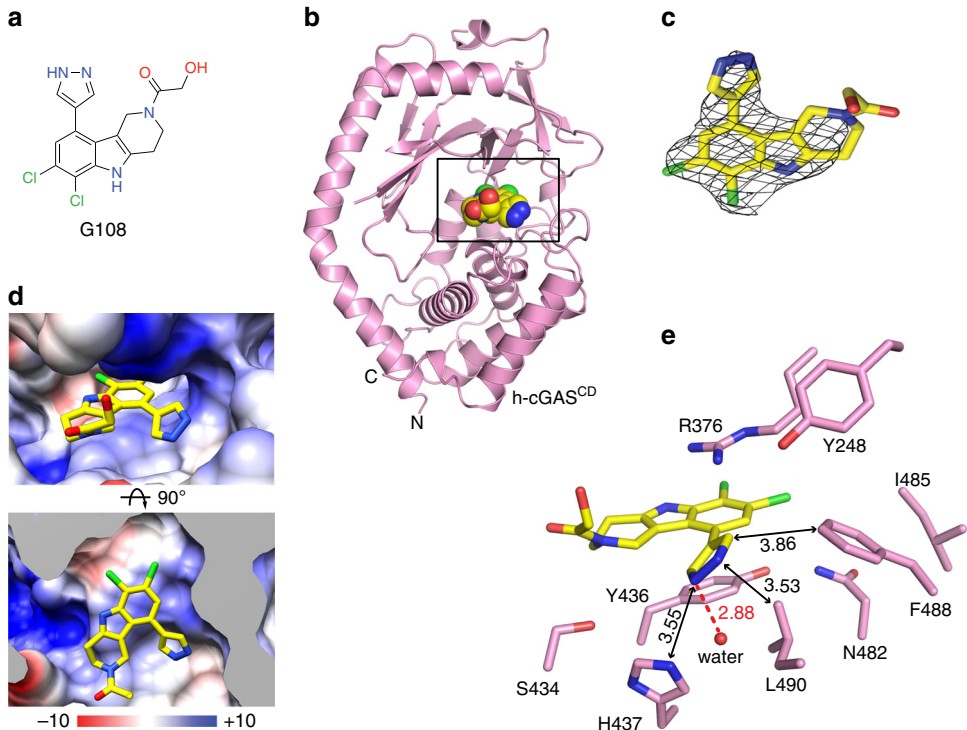

**Fig. 4** Structure of **G108** bound to apo h-cGAS<sup>CD</sup>. **a** Chemical formula of **G108**. **b** Crystal structure of **G108** bound to apo h-cGAS<sup>CD</sup>. The bound **G108** is shown in a space-filling representation, and the binding pocket is boxed. **c** 2Fo–Fc electron density map of bound **G108** contoured at 1.2 σ level. The electron density is poorly defined for the hydroxyl-ethanone side chain attached to the non-planar six-membered ring. **d** Two views of **G108** positioned in its binding pocket within h-cGAS<sup>CD</sup> with the protein shown in an electrostatic surface representation. Electrostatic surface potentials were calculated with Coulombic Surface tool in Chimera with thresholds ±10 kcal mol$^{-1}$ e$^{-1}$. **e** Intermolecular contacts and key distances between **G108** and amino acids lining the binding pocket of h-cGAS<sup>CD</sup>. Distances are in angstrom. Red dashed line indicates hydrogen bond

1.2 σ) for bound **G108** can be traced for the chlorine-substituted pyridoindole core and the pyrazole ring of the bound **G108**, but not for the hydroxyl-ethanone side chain attached to the non-planar six-membered ring, due to its apparent flexibility (Fig. 4c). Expanded top and side views of the bound **G108** in its binding pocket (electrostatic surface potential representation) are shown in Fig. 4d. The intermolecular contacts involving bound **G108** in the catalytic pocket are shown in Fig. 4e, with the inhibitor surrounded by aromatic amino acids Tyr436, His437, and Phe488, hydrophobic amino acids Leu490, and polar amino acids Arg376 and Asn482 (Fig. 4e). The tricyclic pyridoindole core of bound **G108** was sandwiched between the guanidinium group of Arg376 and the aromatic ring of Tyr436, while the pyrazole ring of bound **G108** was anchored in a hydrophobic pocket lined by the side chains of His437, Phe488, and Leu490 of h-cGAS (Fig. 4e). The pyrazole ring of bound **G108** formed a hydrogen bond with one water molecule in the binding pocket. Both chloride atoms of **G108** formed favorable electrostatic interactions within the ligand-binding pocket.

**Structure of the inhibitor G150 bound to apo h-cGAS<sup>CD</sup>.** We also solved the 2.40 Å structure of apo h-cGAS<sup>CD</sup>(K427E/K428E) bound to **G150** (Fig. 5a) using molecular replacement based on the structure of apo h-cGAS<sup>CD</sup> (PDB: 4O68), and refined it to $R_{free}$ of 25.7% and $R_{work}$ of 21.3% with good stereochemistry (X-ray statistics in Supplementary Table 3). The overall structure of the **G150**-h-cGAS<sup>CD</sup>(K427E/K428E) complex is shown in Fig. 5b. The electron density in the 2Fo–Fc map (contoured at 1.2 σ) for bound **G150** can be traced for the chlorine-substituted pyridoindole core and the 2-amino pyridine ring of the bound **G150**, together with part of the hydroxyl-ethanone side chain attached

to the non-planar six-membered ring (Fig. 5c). **G150** was also localized within the ATP and GTP binding pocket (electrostatic representation) as shown in two orientations in Fig. 5d.

We observed differences and similarities in positioning of **G150** relative to **G108**. While the pyridoindole tricyclic core of bound **G150** remained sandwiched between the guanidinium group of Arg376 and the aromatic ring of Tyr436, the side chain of Arg376 adopted a different alignment (Fig. 5e, f). The 2-amino pyridine ring of bound **G150** was anchored in a hydrophobic pocket lined by the side chains of Phe488 and Leu490 of h-cGAS<sup>CD</sup>, yet the side chain of Phe488 was shifted in position to accommodate the larger 2-amino pyridine ring of bound **G150** compared to the pyrazole group of **G108**. Moreover, the ring nitrogen of the 2-amino pyridine ring formed a hydrogen bond with Tyr248, and the hydroxyl-ethanone side chain of **G150** hydrogen-bonded with Ser434 of h-cGAS<sup>CD</sup> (Fig. 5e).

**G108** and **G150** are specific inhibitors of h-cGAS. The drug-binding pocket is lined with conserved amino acids except for Tyr248, Ser434, and Asn482, which are different in mouse. We generated mouse amino acid replacement mutant h-cGAS (N482H) and h-cGAS(Y248F) and measured their enzymatic activity (Supplementary Figure 7a) and drug IC$_{50}$ (Supplementary Figure 7b, c) using RF-MS. h-cGAS(N482H) was enzymatically inactive, indicating that accommodation of the larger histidine side chain requires additional compensatory amino acid changes present in mouse and cannot be studied in isolation. However, while removal of a hydroxyl in h-cGAS(Y248F) retained full activity, it reduced drug inhibition 250-fold for **G108** and **G150**. This was unanticipated considering that **G150** but not **G108** was hydrogen-bonded to the hydroxyl of Tyr248 according to the crystal structures.

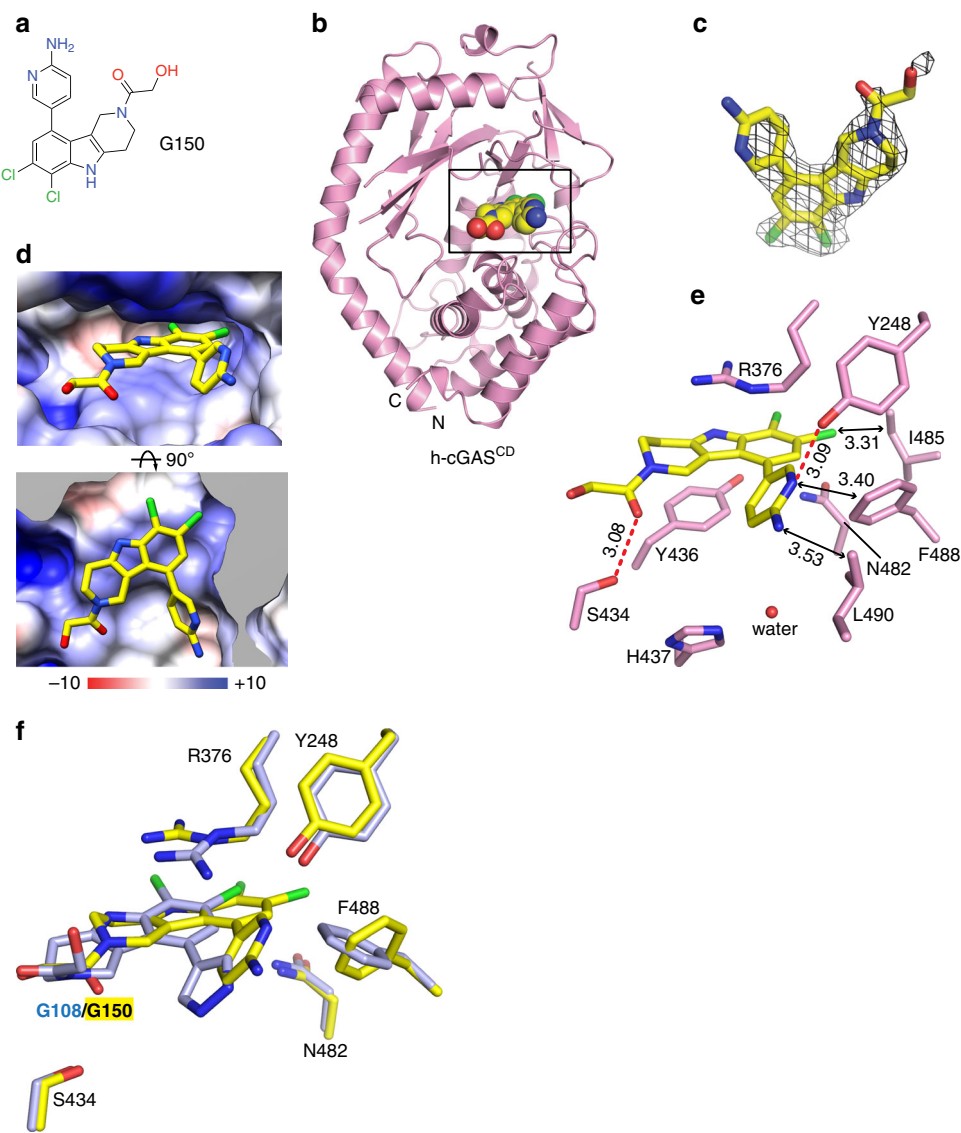

**Fig. 5** Structure of **G150** bound to apo h-cGAS^CD. **a** Chemical formula of **G150**. **b** Crystal structure of **G150** bound to apo h-cGAS^CD. The bound **G150** is shown in a space-filling representation, and the binding pocket is boxed. **c** 2Fo–Fc electron density map of bound **G150** contoured at 1.2 σ level. The electron density is partly defined for the hydroxyl-ethanone side chain attached to the non-planar six-membered ring. **d** Two views of **G150** positioned in its binding pocket within h-cGAS^CD with the protein shown in an electrostatic surface representation. Electrostatic surface potentials were calculated with Coulombic Surface tool in Chimera with thresholds ±10 kcal mol⁻¹ e⁻¹. **e** Intermolecular contacts and key distances between **G150** and amino acids lining the binding pocket of h-cGAS^CD. Distances are in angstrom. Red dashed lines indicate hydrogen bond. **f** Superposition of the structures of **G108** (light blue) and **G150** (yellow) as observed in their complexes with h-cGAS^CD. Note that **G150** is positioned deeper in the binding pocket than **G108** so that the pair of chlorine atoms do not superposition with each other

We have also solved the structure of apo h-cGAS^CD(K427E/ K428E) bound to 2′,3′-cGAMP (2.60 Å resolution and X-ray statistics in Supplementary Table 3) for comparison to bound **G108** and **G150**. Notably, **G108**, **G150**, and 2′,3′-cGAMP targeted the same h-cGAS binding pocket and used similar binding principles involving intercalation of an aromatic element of the ligand between Arg376 and Tyr436 (Supplementary Figure 8). The structural details of 2′,3′-cGAMP bound to apo h-cGAS^CD are summarized in Supplementary Figure 9.

**Potency and specificity of cGAS inhibitors in macrophage cells.** *CGAS* mRNA is well expressed in human monocytic THP1 and mouse macrophage RAW 264.7 cells and the encoded protein readily activated by exposure of cells to dsDNA. The activation of the h-cGAS-STING pathway is accompanied by dramatic *IFNB1*

mRNA induction, which in turn is readily monitored by RNA-sequencing (RNA-seq) or qRT-PCR analysis[4]. We determined the dose-response of **J014**, which showed a biochemical IC₅₀ of 60.0 and 100 nM for m- and h-cGAS, respectively, for blocking the dsDNA-dependent induction of *IFNB1* mRNA using qRT-PCR. Although **J014** inhibited both human THP1 and mouse RAW 264.7 cells with IC₅₀ values of 2.63 μM and 3.58 μM, respectively (Supplementary Figure 10a), we observed the same level of inhibition when we used cGAMP directly rather than dsDNA as activating ligand (Supplementary Figure 10b–e). Cytotoxicity analysis of **J014** in THP1 cells showed a half-maximal cellular lethal dose (LD₅₀) of 28.9 μM (Supplementary Figure 10f). Considering the measurable toxicity and lack of cGAS specificity, we did not further explore this class of compounds.

Cell-based testing of the scaffold **G** compounds was initiated using **G022**, which was a potent inhibitor of cGAS with a

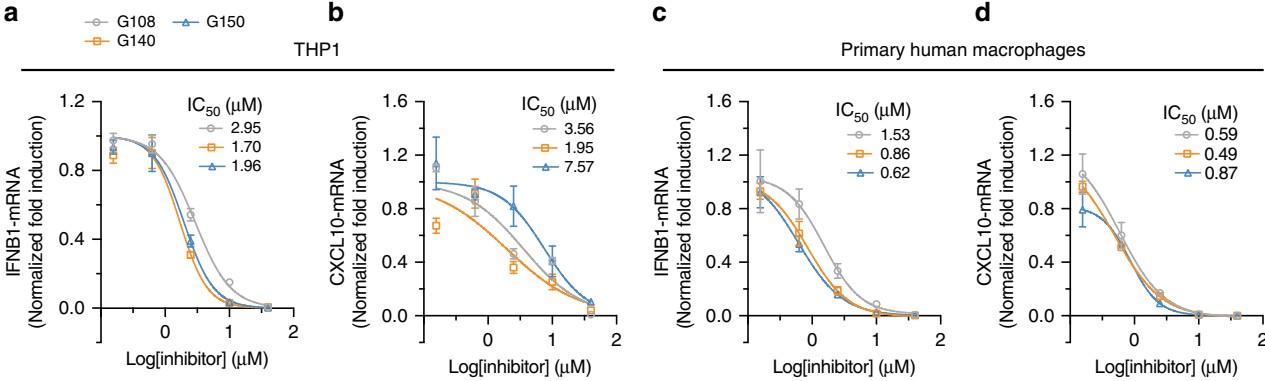

**Fig. 6** G chemotype inhibitors show potent inhibition of cGAS in human THP1 and primary macrophages. **a**, **b** Potency of **G108**, **G140**, and **G150** was tested in dsDNA-stimulated THP1 cells in the presence of a range of different inhibitor concentrations. *IFNB1* (**a**) and *CXCL10* (**b**) mRNA were measured by qRT-PCR for each of the indicated inhibitor concentrations and normalized to no inhibitor control. **c**, **d** Analysis of inhibition of dsDNA-induced cGAS activity by **G108**, **G140**, and **G150** in primary macrophage cells differentiated from human blood-derived monocytes. The cellular IC$_{50}$ values were calculated using GraphPad Prism (7.01).(*n* = 3; mean ± S.D. Data shown are representation of two independent experiments.)

biochemical IC$_{50}$ of 0.018 μM and 0.106 μM for m- and h-cGAS, respectively. **G022** and **G097** inhibited the cGAS pathway in mouse RAW 264.7 cells with a cell-based IC$_{50}$ of 3.32 μM and 1.09 μM, but showed no inhibition in human THP1 cells (Supplementary Figure 11a, b). The **G022** derivatives **G086** and **G092** with cross-species biochemical IC$_{50}$ values of less than 100 nM (Supplementary Figure 6) also did not inhibit *IFNB1* mRNA induction in THP1 cells. Compound **G098**, another strong biochemical inhibitor, could not be tested because of its limited solubility in cell culture media. However, the more potent h-cGAS-specific inhibitors **G108, G140**, and **G150**, showed dose-dependent inhibitory activity in THP1 cells with cellular IC$_{50}$ of 2.95 μM, 1.70 μM, and 1.96 μM, respectively (Fig. 6a), as determined by qRT-PCR for *IFNB1* mRNA. We also determined the drug inhibitory effects on downstream expression of interferon-stimulated genes (ISGs) monitoring *CXCL10* mRNA induction as a surrogate parameter, and observed similar IC$_{50}$ values except for **G150**, which showed about 4-fold reduced potency (Fig. 6b). Furthermore, primary human macrophages derived from human blood yielded IC$_{50}$ values of ≤1 μM (Fig. 6c, d), showing 2- to 8-fold improvement in potency compared to THP1 cells. The cell-based LD$_{50}$ for **G108, G140**, and **G150** were determined to be 56.7 μM, >100 μM, and 53.0 μM, respectively (Supplementary Figure 11c), thereby providing a significant window between efficacy and toxicity for drug testing.

Next, we evaluated the specificity of **G108, G140**, and **G150** for inhibiting dsDNA sensing in THP1 cells. At inhibitor concentrations of 10 μM, representing cellular IC$_{90}$ or more for the inhibitors against cGAS pathway (Fig. 7a), only **G108** showed weak off-target inhibition of the directly cGAMP-stimulated STING pathway and the hairpin-RNA (hpRNA) stimulated RIG-I pathway (Fig. 7b, c). The specificity assay in primary human macrophages using inhibitor concentrations of 5 μM showed 75–95% inhibition of the cGAS pathway (Fig. 7d) without any inhibition of cGAMP-, or hpRNA-stimulated induction of *IFNB1* mRNA (Fig. 7e, f).

We also tested interference of **G108, G140**, and **G150** with other pathways including enzymes not directly implicated in IFNB1 induction[37]. We evaluated the promiscuity toward the unrelated ATP-converting enzyme soluble adenylyl cyclase (ADCY10) in a biochemical assay. **G140** and **G150** showed no inhibition of adenylyl cyclase, however, **G108** showed IC$_{50}$ value of 26.4 μM, which was also 9-times higher compared to an established inhibitor of adenylyl cyclase[38] (Fig. 8a). In another assay, we transfected THP1 and human primary macrophages

with poly(I:C), an established activator of 2′,5′-oligoadenylate synthetase (OAS) leading to activation of RNase L pathway and cellular RNA degradation[39], and observed no interference with poly(I:C)-triggered rRNA degradation in THP1 cells (Fig. 8b) and primary human macrophages (Fig. 8c) at 10 μM inhibitor concentrations. THP1 cells and macrophages mount an ISG expression response upon engagement of their type-I IFN receptors by IFNB1 protein; however, IFNB1 itself is not an ISG. We treated primary macrophages with interferon and observed a 2500- and 80-fold induction of *CXCL10* and *ISG15*, respectively, which was unimpaired by the presence of our inhibitors (Fig. 8d, e). Treatment of human primary macrophages with lipopolysaccharide (LPS) engaging TLR4 induced *IFNB1* and *TNF* transcription 30- and 70-fold, respectively, while dsDNA transfection induced *IFNB1* 1000-fold. The induction of *TNF* mRNA by LPS was unimpaired by the presence of our inhibitors (Fig. 8f).

Recently, cGAS-mediated STING activation has also been implicated in NF-κB pathway activation[40]. There were also reports implicating RIG-I-mediated NF-κB[41] suggesting that downstream TBK1 kinase activation may be responsible for some crossover NF-κB pathway activation, whereas TLR activation directly activates the NF-κB pathway. Co-activation of the NF-κB pathway was tested using THP1-Dual cell lines carrying a secreted embryonic alkaline phosphatase (SEAP) reporter gene responsive to NF-κB activation and a secreted luciferase reporting interferon-induced gene expression. Transfection of dsDNA into THP1-Dual cells induced SEAP, which, however, was 7-fold less than a control LPS-mediated TLR4 activation of NF-κB (Supplementary Figure 11d). Luciferase activity was also induced, but was 10-fold higher with dsDNA compared to LPS addition (Supplementary Figure 11e). The IC$_{50}$ values determined by measuring the inhibition of the SEAP NF-κB pathway reporter (2.27 μM, 1.36 μM, and 0.90 μM for **G108, G140**, and **G150**, respectively) agreed with those of the luciferase reporter monitoring the IFNB1 response (Supplementary Figure 11f, g).

## Discussion

We developed an ATP-coupled high-throughput LUM assay to identify small-molecule inhibitors of h-cGAS and screened nearly 300,000 library compounds. The LUM assay was faster and required fewer resources in comparison to our previously used RF-MS assay identifying inhibitors of m-cGAS. The LUM assay

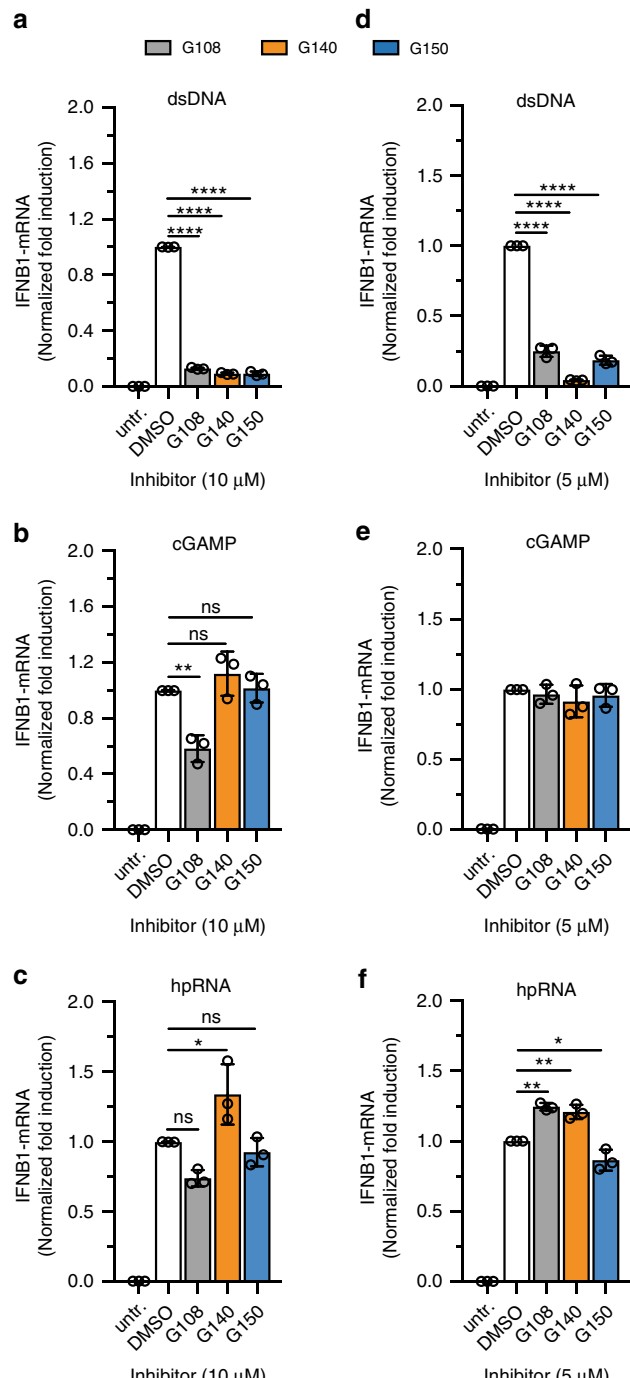

**Fig. 7** G chemotype inhibitors show specific inhibition against cGAS in human THP1 and primary human macrophages. **a–c** Specificity of **G108**, **G140**, and **G150** against cGAS activity was tested using different ligands [2 µg ml⁻¹ dsDNA (cGAS), 10 µg ml⁻¹ cGAMP (STING), 0.5 µg ml⁻¹ hpRNA (RIG-I)] for *IFNB1* mRNA induction followed by any possible inhibition in the presence of 10 µM inhibitor concentration in THP1 cells. **d–f** Specificity analysis of 5 µM **G108, G140**, and **G150** against cGAS inhibition in primary human macrophage cells differentiated from human blood-derived monocytes using different ligands: 2 µg ml⁻¹ dsDNA (cGAS), 10 µg ml⁻¹ cGAMP (STING), or 0.5 µg ml⁻¹ hairpin RNA (hpRNA) (RIG-I). Data shown are mean ± S.D. for three technical replicates and are representation of two independent experiments ($n = 3$; mean ± S.D.; *$p = 0.04$, **$p < 0.01$, ****$p < 0.0001$ using one-way ANOVA followed by Tukey's test for multiple comparison (**a–d**, **f**). Data shown are representation of two independent experiments.)

was also reliable considering that 20 out of 27 hit compounds were confirmed in the secondary RF-MS assay using h-cGAS.

Prioritizing two cross-species active hit compounds from the screen, **J001** and **G001**, we launched a comprehensive medicinal chemistry program optimizing these inhibitors including cell-based assays for potency and specificity. This process yielded **J014**, a phenyl-substituent-containing derivative of **J001**, with a 10-fold improved biochemical inhibitory activity for both h- and m-cGAS relative to its parent compound. **J014** inhibited dsDNA-induced IFNB1 transcription in mouse and human macrophage cell lines; however, it also unexpectedly inhibited directly cGAMP-induced *IFNB1* transcription, suggesting unspecific interference with transcriptional pathways. We do not yet understand the off-target mechanism but excluded that **J014** acted as dsDNA intercalator[32] (Supplementary Data 1). Additionally, considering the difficulty of replacing the nitro group without substantial loss in the biochemical IC₅₀ value, the **J** scaffold is facing substantial hurdles for further drug development. The derivatization of **G001** resulted in potent biochemically cross-species active compounds with IC₅₀ values in the tens of nanomolar range, representing a 20-fold improved potency over the parent compound. Surprisingly, most of the biochemically cross-species active drugs were only active in mouse macrophage cells, either reflecting differences in kinetics of uptake or drug-metabolizing enzymes between mouse and human macrophages and/or differences in the mechanisms of m- and h-cGAS activation. Only the most potent h-cGAS-specific derivatives with added pyrazole (**G108**), methyl-pyrazole (**G140**), or 2-amino pyridine (**G150**) moieties showed inhibitory activity in THP1 cells and primary human macrophages. **G140** and **G150** showed a complete absence of off-target effect on a diverse range of sensors, while **G108** displayed a modest 20–40% inhibition of the directly cGAMP stimulated STING pathway and hpRNA stimulated RIG-I pathway at 10 µM inhibitor concentration in THP1 cells. These compounds also show no substantial cellular toxicity. Therefore, the **G** class of compounds, especially **G140** and **G150**, are promising for h-cGAS drug development, and also yielded m-cGAS inhibitors more potent than our previously identified **RU.521**.

The cellular IC₅₀ values of cGAS inhibitors were almost 100-fold higher compared to the biochemical assay IC₅₀ values. Such discrepancy between cellular and biochemical inhibitory potency suggests that membrane barriers need to be overcome reducing effective intracellular concentration and/or the drug may be metabolized or transported out of cells. Consequently, the ranking of compounds by biochemical IC₅₀ values did not always predict the compounds with the best potency and selectivity in the cell-based assays.

We were able to obtain co-crystal structures of apo-h-cGAS^CD with **G150**, one of our best h-cGAS inhibitors, and **G108**, the analog with some off-target activity, and localizing them in the reactive ATP- and GTP-binding pocket. These compounds were poor inhibitors of m-cGAS with biochemical IC₅₀ values ≥5 µM. It is compelling to speculate about the underlying differences in amino acids lining the binding pockets in h- and m-cGAS. Notably, the flexible side chain of Asn482 in h-cGAS^CD is substituted by the more rigid planar ring of His467 at the equivalent position in m-cGAS^CD. However, Asn482His substitution in h-cGAS^CD resulted in complete loss of enzymatic activity and the effect on **G108** or **G150** could not be assessed. Another relevant difference is the presence of the hydroxyl group on Tyr248 in h-cGAS^CD, which is absent at the equivalent position in m-cGAS^CD corresponding to Phe488, but was hydrogen-bonded to the 2-amino pyridine ring of **G150**. Here, the substitution Tyr248Phe in h-cGAS^CD did not compromise the enzyme activity and resulted in a strong, anticipated reduction in inhibitory activity of **G150**.

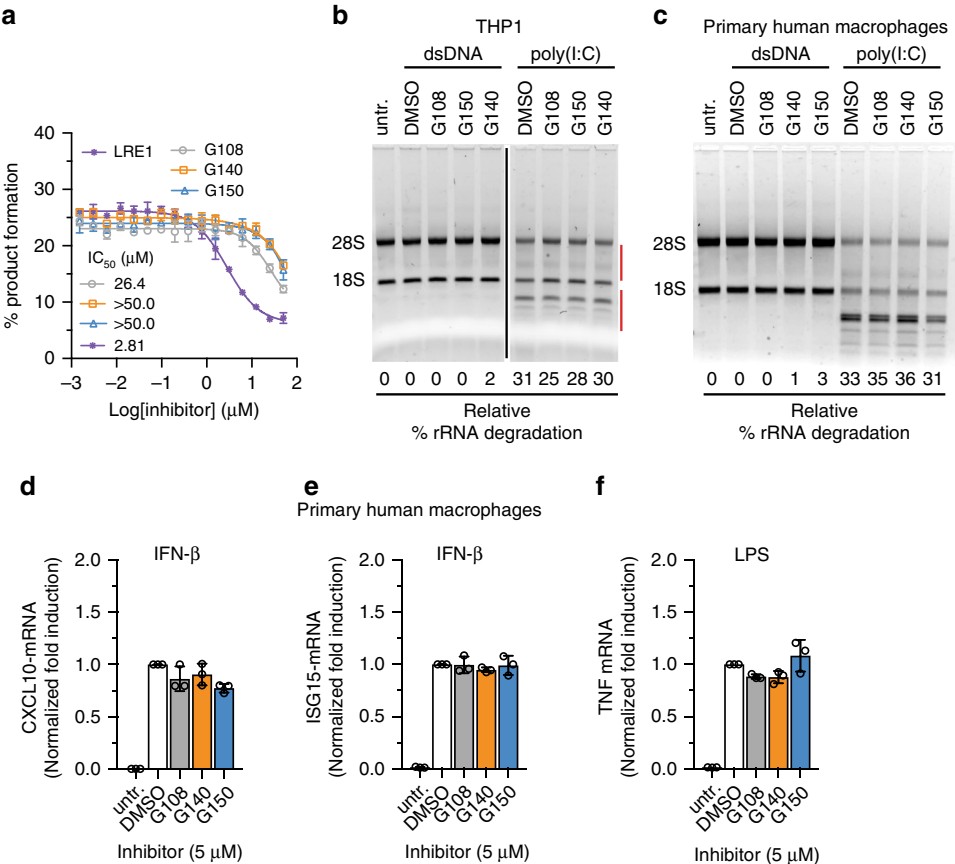

**Fig. 8** G chemotype inhibitors show no off-target effect against other nucleotidyl transferase enzymes. **a** Biochemical off-target assessment of cGAS inhibitors using soluble adenylyl cyclase enzyme. **b, c** Cellular off-target effect assessment of cGAS inhibitors against OAS proteins using 2 μg ml$^{-1}$ poly (I:C) as a ligand to stimulate OAS-RNaseL pathway in THP1 cells (**b**) and primary human macrophages (**c**). Red bars encompass the degraded rRNAs that were used for the quantification of relative percentage (%) rRNA degradation. **d–f** Off-target analysis of 5 μM **G108, G140**, and **G150** against cGAS inhibition in primary human macrophage cells differentiated from human blood-derived monocytes using different ligands: 1000 U ml$^{-1}$ recombinant human interferon-β (IFN-β) (JAK/STAT), or 2 μg ml$^{-1}$ LPS (TLR4) (n = 3; mean ± S.D. Data shown are representation of two independent experiments (**a**, **b**, **d**–**f**).)

However, we also observed a similar reduction for **G108**, which although it has the ability of hydrogen-bond by its pyrazole substituent was not as deeply inserted into the binding pockets of h-cGAS$^{CD}$ to reach the Tyr248. Considering that the **G** scaffold compounds do not fully occupy the ATP and GTP binding pocket of cGAS and have flexibility to reposition in the large pocket while maintaining the important aromatic stacking interaction, molecular dynamics modeling remains challenging due to limited steric constraints and factoring in of water and metal ion networks. For the same reasons, it is also challenging to reconcile structurally the inhibitor optimization pathway.

In summary, we developed a class of h-cGAS specific and potent small-molecule inhibitors active in primary human macrophages. These compounds may have utility for development of therapeutic treatment of cGAS-related human diseases and will be very useful tools in furthering our understanding of the role of the cGAS pathway in pathological processes.

## Methods

**Chemical libraries**. Library screening compounds were purchased from different vendors at different times over a period of 10 years using various compound selection strategies depending on factors such as cost per compound, budget, and diversity and drug-likeness (for example, Lipinski Filters, PAINS filters, QED scores)[34]. Compound vendors were contractually required to provide at least 85% purity, and provide LC-MS and NMR certificates on every compound. In some cases, the libraries were sold as existing sets with no option for removing unde-sirable substructures, and in other cases, the vendor offered the opportunity to

select a subset of its very large commercially available compound collection. In cases where we had the option to select a subset of vendor compounds, we used the Biovia pipeline pilot software (http://accelrys.com/products/collaborative-science/biovia-pipeline-pilot). For example, we often selected a diverse number of compounds based on a yearly budget. We did this by filtering out the PAINS, non-Lipinski-like compounds and compounds similar to the existing collection from vendor files by using Tanimoto similarity calculations[42] and then applying fingerprint-based (FCFP6) clustering algorithms, which parsed the compounds into groups of similar structures. The average group size was set to a number of groups that we could afford to purchase in a particular time, and only those compounds positioned in the center of clusters were purchased in order to max-imize diversity. Supplementary Table 4 shows the number of compounds pur-chased from each vendor, and the smiles string of every compound screened is included along with the screening results deposited into NCBI screening library database. A total of 281,348 unique compounds were studied in the primary screen. Compounds were dissolved in DMSO to 5 mM in 96-well tube arrays and then partitioned into ten aliquots of 20 μl each and two master copies of 80 μl each into 384-well polypropylene deep small volume microplate (Greiner Bio-One), which were stored at −28 °C. Only a single 20-μl copy was used as a compound source plate from which to draw compound into the assay plate, to a maximum of 20 freeze–thaw cycles.

**dsDNA preparation for cGAS activity assay**. Sense (s) and antisense (as) 45-nt (s, 5′-TACAGATCTACTAGTGATCTATGACTGATCTGTACATGATCTACA-3′; as, 5′-TGTAGATCATGTACAGATCAGTCATAGATCACTAGTAGATCTGT A-3′)[32] and 100-nt oligonucleotides (s, 5′-ACATCTAGTACACATGTCTAGTCAGT ATCTAGTGATTATCTAGACATACATCTAGTACACATGTCTAGTCAGTATCTA GTGATTATCTAGACATGGACTCATCC-3′; as, 5′-GGATGAGTCCATGTCT AGTAATAACTAGTACTGACTAGACACATGTACTAGATGTATGTCTAGAT AATCACTAGTACTGACTAGACATG TACTAGTGT-3′)[27] were purchased at 1 μmol scale from IDT and dissolved in 150 mM NaCl, 1 mM dithiothreitol (DTT),

20 mM Tris-HCl, pH 7.5, annealing buffer to obtain a 1-mM single-stranded DNA stock solution. The quality of oligonucleotides was verified by denaturing polyacrylamide gel electrophoresis. Annealing of s and as strands was conducted by incubating 200 μM each of s and as strand in annealing buffer at 95 °C for 90 s followed by slow cooling to 25 °C at a rate of 0.1 °C s$^{-1}$ in a Peltier thermocycler (BioRad). The completion of annealing was verified by native agarose gel electrophoresis[32]. A modified pUT7 plasmid corresponding to 3200-bp DNA was isolated from transformed *Escherichia coli* Top10 with a PureLink HiPure Plasmid Megaprep Kit (Invitrogen). The plasmid was linearized with HindIII (NEB) and further purified by standard ethanol precipitation method and resuspended in MilliQ water. Herring testes DNA (HT-DNA) was purchased from Sigma (catalog number: D6898).

**Recombinant cGAS protein expression and purification.** Full-length mouse cGAS (m-cGAS) has been described elsewhere[32]. The gene encoding human cGAS (h-cGAS) was purchased from Open Biosystems Inc (catalog number: 115004). The sequence corresponding to full-length human cGAS was inserted into a modified pRSFDuet-1 vector (Novagen) such that an ubiquitin-like protease (ULP1) cleavage site separated cGAS from the preceding hexahistidine (His$_6$)-SUMO tag. The sequence was subsequently confirmed by sequencing. The His$_6$-SUMO tagged protein was expressed in *E. coli* strain BL21-CodonPlus(DE3)-RIL (Stratagene, Santa Clara, CA, USA). Bacteria were cultured at 37 °C to an OD$_{600}$ of 0.8, at which point the culture temperature was reduced to 18 °C and isopropyl β-D-1-thiogalactopyranoside (IPTG) was added to a final concentration of 0.3 mM for induction overnight. Bacteria were harvested by centrifugation at 4 °C and disrupted by sonication in buffer W (500 mM NaCl, 20 mM imidazole, and 50 mM Tris-HCl, pH 7.5) supplemented with 1 mM phenylmethylsulfonyl fluoride (PMSF) protease inhibitor and 5 mM β-mercaptoethanol. Cell lysates were centrifuged at 20,000 rpm (47,850 × *g*) for 1 h in a JA-20 fixed angle rotor (Avanti J-E series centrifuge, Beckman Coulter). After centrifugation, the supernatant was loaded onto a HisTrap FF column (GE Healthcare, Little Chalfont, Buckinghamshire, UK) and extensively washed with buffer W. Thereafter, the target protein was eluted with buffer W supplemented with 500 mM imidazole. The His$_6$-SUMO tag was cleaved and removed by ULP1 during dialysis at 4 °C overnight against buffer containing 300 mM NaCl, 1 mM DTT, 20 mM Tris-HCl, pH 7.5, and 5% glycerol. After dialysis, the tags were removed by chromatography using a HisTrap FF column. The cGAS protein containing flow-through was further fractionated over a Heparin column (GE Healthcare), followed by gel filtration on a 16/60 G200 Superdex column (GE Healthcare) in buffer G (300 mM NaCl, and 1 mM DTT, 20 mM Tris-HCl, pH 7.5). The same protocol was used for the purification of the wild-type h-cGAS$^{CD}$ (152–522) and mutant h-cGAS$^{CD}$ (K427E/K428E). All the plasmids [pRSFDuet-sumo-h-cGAS$^{CD}$(K427E/K428E), pRSFDuet-sumo-m-cGAS, and pRSFDuet-sumo-h-cGAS] are deposited at http://www.addgene.org.

**High-throughput screening.** Inhibitor primary compound screening was carried out in 20 μl final volumes in 384-well flat bottom polystyrene microplates (Lumitrac) by following any addition of reagents with a 30-s centrifugation at 180 × *g* to ensure that all liquid was collected at the bottom of the well. The final concentration of full-length h-cGAS, dsDNA, ATP, and GTP were 100 nM, 25 nM, 100 μM, and 100 μM, respectively. A total of 10 μl of 1× reaction buffer composed of 20 mM Tris-HCl, pH 7.4, 150 mM NaCl, 5 mM MgCl$_2$, 1 μM ZnCl$_2$, and 0.01% Tween-20 per well was dispensed using a Thermo Multidrop Combi dispenser (Thermo Scientific). 0.05 μl of 5 mM stock compounds in DMSO were dispensed into assay microplates with a Janus 384 MDT NanoHead (PerkinElmer). Final concentration of the screening compounds in the assay was 12.5 μM. A concentration of 0.5% DMSO did not interfere with cGAMP production from recombinant cGAS. Plates were sealed using a Velocity11 PlateLoc thermal plate sealer. Plates were stored at 4 °C overnight. The next morning 5 μl of a 4× master mix of 0.4 mM ATP, 0.4 mM GTP, 0.1 μM dsDNA, prepared in 1× reaction buffer supplemented with 2 mM DTT, was added to wells in columns 1–23 using a Thermo Multidrop Combi dispenser, while 5 μl of the same master mix without dsDNA (control for no enzymatic activity) was added to wells in column 24. The reaction was started by adding 5 μl of a 4× h-cGAS solution of 0.4 μM prepared in 1× reaction buffer supplemented with 2 mM DTT to each and incubation for 7 h at RT, with the plates sealed. The reaction was stopped by addition of 20 μl of Kinase-Glo® Max Luminescent Kinase Assay (Promega, Madison, WI). The luminescence was recorded in relative light units (RLUs) using a Biotek Synergy Neo plate reader (BioTek, Winooski, VT). The ATP depletion was normalized against the positive control (column 24) and negative control (column 23) as follows: % inhibition = 100 × (RLU$_{sample}$−RLU$_{average\ negative\ control}$)/(RLU$_{average\ positive\ control}$−RLU$_{average\ negative\ control}$). Compounds that inhibited cGAS activity by ≥40% were retested by a concentration response experiments to determine the half maximal inhibitory concentration (IC$_{50}$). For this, stock compounds at 5 mM were serially diluted in half in DMSO for a total of ten dilutions. The same assay protocol as described above was used for validation of hit compounds but with the following differences: 0.1 μl of serially diluted compounds were dispensed with a Janus 384 MDT NanoHead (PerkinElmer) and the assay was performed 1 h after the addition of compounds. The highest final assay compound concentration was 25 μM. Subsequently, any compound selected for follow-up studies was reordered from available vending sources (Supplementary Table 2), dissolved in DMSO at 10 mM and re-tested in concentration response experiments. The IC$_{50}$ values for the

enzymatic assay were calculated using GraphPad Prism (7.01) from three replicate experiments. The values represent mean IC$_{50}$±S.D. The quality of the screen was assessed by Z′ factor[43].

**dsDNA intercalation assay.** dsDNA intercalation was assessed for validated hit compounds using a high-throughput fluorescence polarization (FP) assay following the protocol developed at the Broad Institute (National Center for Biotechnology Information. PubChem BioAssay Database; AID = 504727, https://pubchem.ncbi.nlm.nih.gov/bioassay/504727). The assay was performed in 384-solid bottom opaque plates in a final volume of 30 μl. A total of 10 μl of HEN buffer (10 mM HEPES pH 7.5, 1 mM EDTA pH 7.5, 100 mM NaCl) was dispensed per well using a Thermo Multidrop combi dispenser (Thermo Scientific); 0.18 μl of compounds dissolved in DMSO at 5 mM stock concentration were dispensed per well with a Janus 384 MDT NanoHead (Perkin Elmer). Mitoxantrone at 50 μM final concentration was used as a positive control while DMSO alone was used as a negative control. Ten microliters of a solution of 150 nM acridine orange in HEN buffer was dispensed per well using a Thermo Multidrop Combi dispenser (Thermo Scientific). Finally, 10 μl of a solution of 45-bp dsDNA at 37.5 μg ml$^{-1}$ was dispensed per well using a Thermo Multidrop combi dispenser (Thermo Scientific). The liquid was collected at the bottom of well by centrifugation at 180 × *g* for 30 s and the plates were incubated for 30 min. Biotek Synergy Neo plate reader was used for the measurement of FP. Samples were excited at 485 nm and fluorescence emission was detected through a 530 nm filter, taking ten measurements per data point. All FP values expressed in millipolarization (mP) units were calculated using the equation mP = 1000 × [($I_S$−$I_{SB}$)−($I_P$−$I_{PB}$)]/[($I_S$−$I_{SB}$) + ($I_P$−$I_{PB}$)], where $I_S$ and $I_P$ refer to the parallel and perpendicular emission intensity, respectively, and $I_{SB}$ and $I_{SP}$ corresponds to parallel emission intensity and perpendicular emission intensity of the buffer, respectively. The experimental *G*-factor, defined as $G = I_S/I_P$, was set to 1.0 in all experiments.

**RapidFire 365 mass spectrometry (RF-MS) validation assay.** Freshly sourced powder of hit compounds or in-house synthesized analogs of hit compounds were resuspended in 10 mM stock concentration using DMSO and tested to determine their IC$_{50}$ values against h-cGAS and m-cGAS using RapidFire 365 mass spectrometry (RF-MS). Reaction solutions of 20 μl were incubated for 2 h (m-cGAS) or 7 h (h-cGAS) at RT and stopped by addition of 60 μl of 0.5% (v/v) formic acid per well followed by RF-MS analysis. An aqueous solvent of 5 mM ammonium acetate, pH 10 was used for loading/washing process. An organic solvent comprising 5 mM ammonium acetate, pH 10 in 50% water, 25% acetone, and 25% acetonitrile was used for elution of the analytes. About 35 μl of each sample was aspirated from a 384-well plate and separated using a Graphitic carbon Type D cartridge. The sample loaded onto cartridge was then washed for 4 s at 1.5 ml min$^{-1}$ using the aqueous solvent. ATP, GTP, and cGAMP were eluted for 5 s using the organic solvent at a flow rate of 1.5 ml min$^{-1}$ followed by re-equilibration with the aqueous solvent for 5 s at a flow rate of 1.5 ml min$^{-1}$. The samples were analyzed using a negative ionization mode in the mass spectrometer, with a gas temperature of 350 °C, nebulizer pressure of 35 psi, and gas flow rate of 15 l min$^{-1}$. The acquisition range was between 300 and 800 m/*z* for all the chromatograms and the molecular masses of the detected peaks were: ATP: 505.9835, GTP: 521.9854, and cGAMP: 673.0906. The area under the curve (AUC) of the extracted ion counts for each analyte was calculated using the Agilent RapidFire Integrator software. Percent product formation for cGAMP was calculated as: product formation (%) = [(AUC$_{cGAMP}$ × 100)/(AUC$_{cGAMP}$ + ½ AUC$_{ATP}$ + ½ AUC$_{GTP}$)]. Percent product formation from each sample at a given inhibitor concentration was used to determine percent inhibition by normalization against the positive control (column 24) and negative control (column 23). The % inhibition was calculated as follows: % inhibition = 100 × [(sample-average negative control)/(average positive control − average negative control)].

**Crystallization.** In order to improve the stability and crystallization behavior of h-cGAS$^{CD}$ (152–522), we generated the K427E/K428E double mutant. This improved the outcome of crystal soaking at high concentration of DMSO and ligands. To assemble the h-cGAS$^{CD}$-ligand binary complex, purified h-cGAS$^{CD}$(K427E/K428E) in buffer (300 mM NaCl, and 1 mM DTT, 20 mM Tris-HCl, pH 7.5) was mixed with 10% (v/v) samples of excess ligand dissolved either in aqueous buffer (cGAMP) or DMSO (**G108** and **G150**). The mixtures were then slowly shaken for 2 h at 4 °C. The samples of the complexes were next centrifuged at 13,000 × *g* to remove precipitants right before crystallization. Sample concentrations were approximately 1 mM in ligand and 150 μM in h-cGAS$^{CD}$(K427E/K428E). Crystallization screens were next undertaken using a MOSQUITO robot and hits optimized using the hanging drop vapor diffusion method at 20 °C. All crystals were grown in drops mixed from 1 μl of complex solution and 1 μl of reservoir solution (0.064 M sodium citrate 7.0, 0.1 M HEPES, pH 7.0, 10% PEG5000MME for h-cGAS-**G108** and h-cGAS-**G150**, 0.1 M bicine 8.5, 10% PEG6000, pH 9.0 for h-cGAS-cGAMP). In the case of the h-cGAS-**G108** complex, the crystals were additionally soaked in the reservoir solution containing 30 mM **G108** and 30% DMSO for 12 h. In the case of the h-cGAS-**G150** complex, the crystals were additionally soaked in the reservoir solution containing 40 mM **G150** and 40% DMSO for 6 h. We were also able to grow crystals of apo h-cGAS$^{CD}$ with various

other G and J scaffold derivatives but failed to arrive at structures for the following reasons: **G140** co-crystals diffracted poorly and **J014** could not be accurately traced in the binding pocket, perhaps due to poor occupancy in co-crystals.

**Data collection and refinement**. For data collection, crystals were cryoprotected in reservoir solution supplemented with 30% glycerol and flash frozen in liquid nitrogen. The diffraction data sets for h-cGAS$^{CD}$-**G108**, h-cGAS$^{CD}$-**G150**, and h-cGAS$^{CD}$-cGAMP complexes were collected on the 24-ID beamline at the Advanced Photon Source (APS) at the Argonne National Laboratory. All the diffraction data were indexed, integrated, and scaled using the HKL2000 program. The complex structures were solved using molecular replacement method with program PHA-SER[44] of the PHENIX package[45], using the modified apo h-cGAS$^{CD}$ structure (RCSB code: 4O68) as the search model. Model building and structural refinement were carried out using COOT[46] and phenix.refine[47], respectively. The statistics of the data collection and refinement are shown in Supplementary Table 3. Figures were generated using PyMOL (http://www.pymol.org) and Chimera[48].

**Cell culture**. The human acute monocytic leukemia cell line (THP1) (ATCC) was cultured in RPMI 1640 (Gibco) supplemented with 10% heat-inactivated fetal bovine serum (FBS) (Sigma), 100 U/ml penicillin, 100 µg ml$^{-1}$ streptomycin, and 2 mM L-glutamine (referred to as RPMI culture medium). THP1-Dual cells (InvivoGen) were cultured and maintained in RPMI 1640 (Gibco) supplemented with 10% FBS, 50 U ml$^{-1}$ penicillin, 50 µg ml$^{-1}$ streptomycin, 100 µg ml$^{-1}$ normocin, and 2 mM L-glutamine (referred to as RPMI-Luc culture medium). To maintain luciferase expression, 100 µg ml$^{-1}$ of zeocin and 10 µg/ml$^{-1}$ of blasticidin were added to the growth medium every other passage.

Isolation of macrophages from human blood was performed following relevant ethical regulation according to the Rockefeller University Institutional Review Board-approved study protocol. Human monocyte-derived macrophages (MDMs) were isolated from peripheral blood mononuclear cells (PBMCs) isolated from blood of healthy donors obtained in the form of deidentified Leukopak samples from New York Blood Center by adhesion to standard plastic cell culture dishes. Non-adherent cells were washed out after 3 h, with subsequent maintenance of adherent cells in RPMI medium supplemented with 10% fetal calf serum (FCS) and GM-CSF (100 ng ml$^{-1}$) for 3 days to differentiate monocytes to macrophages. The cells were further maintained for 2–3 days in RPMI medium supplemented with 10% FCS before being used for *IFNB1, CXCL10, ISG15*, and *TNF* mRNA induction analysis.

The mouse macrophage cell line (RAW 264.7) (ATCC) was cultured in Dulbecco's modified Eagle's medium (DMEM) supplemented with 10% FBS, 100 U ml$^{-1}$ penicillin, 100 µg ml$^{-1}$ streptomycin, 2 mM L-glutamine, and 110 mg l$^{-1}$ sodium pyruvate (referred to as DMEM culture media). RAW-Lucia cells (InvivoGen) were cultured and maintained in DMEM supplemented with 10% FBS, 50 U ml$^{-1}$ penicillin, 50 µg ml$^{-1}$ streptomycin, 100 µg ml$^{-1}$ normocin, 2 mM L-glutamine, and 110 mg l$^{-1}$ sodium pyruvate (referred to as DMEM-Luc culture media). To maintain expression of luciferase, 200 µg ml$^{-1}$ of zeocin was added to the growth medium every other passage. e-Myco mycoplasma PCR detection kit (Bulldog Bio) was used to routinely check for mycoplasma contamination in all cell lines.

**Soluble adenylyl cyclase (sAC) inhibition assay**. In vitro inhibition of soluble adenylyl cyclase activity was performed in 20 µl final volumes in 384-well poly-propylene plates. Briefly, serially diluted compounds in 10 µl of reaction buffer composed of 100 mM Tris-HCl, pH 7.5, 20 mM MgCl$_2$, 2 mM CaCl$_2$, 40 mM NaHCO$_3$, and 1 mM DTT were added with 5 µl of human sAC$_t$ protein[38]. Reaction was started by adding 5 µl of 4 mM ATP into each well. The samples were incubated at RT for 3 h and stopped by adding 50 µl of 1% formic acid per well. The formation of cAMP and corresponding consumption of ATP for each sample in comparison to standards of known concentrations were determined using RF-MS. A total of 35 µl of sample was aspirated from each well and injected onto a gra-phitized carbon SPE column extraction cartridge, washed with aqueous alkaline buffer, eluted with an alkaline/organic solvent and loaded onto the electrospray-MS to collect mass spectra of each sample. Between each sample injection, the RapidFire sipper was washed using organic (25% acetonitrile, 25% acetone in 5 mM ammonium acetate, pH 10) and aqueous (5 mM ammonium acetate, pH 10) solvents. The mass spectrometry data were processed and analyzed using Agilent MassHunter software. Compound LRE1[38] was used as a positive control. The IC$_{50}$ values were calculated using GraphPad Prism (7.01), from three replicate experiments; the error bars represent SEM.

**rRNA cleavage assay**. Cells preincubated with cGAS inhibitors for 1 h were transfected with 2 µg ml$^{-1}$ of poly(I:C) (InvivoGen) for 8 h. Cells were then harvested and total RNA was extracted using Trizol (Ambion) following manufacturer's instructions. A total of 500 ng of total RNA from each sample was run on 1% agarose gel and the percentage (%) of degraded rRNA was determined by quanti-fying the integrated intensity of different bands on gel using ImageJ software. % degraded rRNA = Intensity$_{degraded\ rRNA}$ / (Intensity$_{degraded\ rRNA}$ + Intensity$_{28S\ rRNA}$ + Intensity$_{18S\ rRNA}$) ×100. Relative % rRNA degradation was determined with respect to % rRNA degradation for untreated sample, i.e., Relative % rRNA degradation = % degraded RNA$_{sample}$ − % degraded RNA$_{untr}$.

**Cytotoxicity assay**. The cytotoxicity of compounds was tested using the CellTiter-Glo luminescent cell viability assay (Promega), which determines the number of viable cells based on ATP concentration upon cell lysis. A source plate with 2-fold serially diluted compounds was prepared using DMSO in a 384-well polypropylene plate starting with 10 mM for the highest compound concentration. The assay was performed in a final volume of 50 µl in 384-transparent bottom white opaque plates. The assay plates were prefilled with 9.5 µl of medium per well using a Thermo Multidrop Combi dispenser. 0.5 µl of the serially diluted compounds from the source plate were transferred to the assay plate using Janus 384 MDT Nano-Head (Perkin Elmer). 40 µl of cell suspensions at $3.75 \times 10^5$ cells ml$^{-1}$ for THP1 and $1.25 \times 10^5$ cells ml$^{-1}$ for RAW 264.7 were transferred into each well of the assay plate using a ThermoMultidrop Combi dispenser. Tamoxifen at 50 µM final concentration and DMSO (vehicle) at 1% final concentration were used as positive and negative control, respectively. Following compound addition, plates were incubated for 24 h in a cell culture incubator. 10 µl of CellTiter-Glo reagent was added to each well using ThermoMultidrop Combi and incubated on orbital shaker for 30 min at RT to induce cell lysis and stabilize luminescent signal. Luminescence (Lum) in each well was recorded using a Biotek Synergy Neo plate reader (BioTek, Winooski, VT). Cell viability for each compound was calculated using vehicle (DMSO) as 100% and Tamoxifen as 0%, i.e., % Viability$_{sample}$ = [(Lum$_{Sample}$ − Lum$_{Tamoxifen}$)/(Lum$_{DMSO}$ − Lum$_{Tamoxifen}$)] × 100.

**Cell-based Lucia luciferase and SEAP assays**. THP1-Dual cells were pre-incubated in 24-well plates ($2.5 \times 10^5$ cells/well, 500 µl per well) over an indicated concentration range of inhibitors for 1 h. DMSO was added as negative control. Cells were transfected with 0.5 µg ml$^{-1}$ of dsDNA ligands of different lengths or herring testes DNA (HT-DNA) in complex with Lipofectamine 2000 (Invitrogen) for 24 h. Transfection complex was prepared by combining 0.25 µg of dsDNA in 25 µl Opti-MEM (Gibco) with 0.25 µl of Lipofectamine 2000 in 25 µl Opti-MEM and adding the 50 µl combined volume for each well containing cells. Luciferase luminescence was measured for each sample using QUANTI-Luc luciferase reagent (InvivoGen) fol-lowing the manufacturer's protocol. Shortly, 20 µl of cell culture supernatant per well was transferred into a 96-well white opaque plate and luminescence was recorded using a Biotek Synergy Neo plate reader (BioTek, Winooski, VT) with the following parameters: 50 µl of luciferase reagent injection, end-point measurement with 4 s start time, and 0.1 s reading time. Luciferase-based reporter assays using mouse RAW-Lucia cells followed a similar protocol except that $1.25 \times 10^5$ cells were pre-cultured for 24 h in 24-well plates prior to addition of inhibitor compounds. In parallel, to perform cGAS-dependent NF-κB pathway inhibition analysis, 50 µl of cell culture supernatant per well was transferred into a 24-well cell culture plate and added with 450 µl of QUANTI-Blue SEAP detection reagent (InvivoGen). The samples were then incubated at 37 °C for 1 h and absorbance (Abs) was measured at 640 nm using Biotek Synergy Neo plate reader (BioTek, Winooski, VT). Relative luciferase activity or relative SEAP activity for each compound-treated sample was calculated using Lipofectamine 2000-treated sample as negative control and Lipofectamine 2000:dsDNA complex-treated sample without compound as positive control, i.e., relative luciferase activity = (RLU$_{sample}$ − RLU$_{negative\ control}$)/(RLU$_{positive\ control}$ − RLU$_{negative\ control}$), where RLU indicates raw luciferase unit, or relative SEAP activity = (Abs$_{sample}$ − Abs$_{negative\ control}$)/ (Abs$_{positive\ control}$ − Abs$_{negative\ control}$). Substituting the 100-bp dsDNA ligand with the same amount of longer dsDNAs (a linearized 3200-bp plasmid dsDNA or herring testes DNA) did not impact IC$_{50}$ values (Supplementary Figure 10h).

**IFNB1 and other mRNA expression analysis of macrophage cells**. Cellular activation of cGAS enzyme leads to *IFNB1* mRNA expression in THP1, RAW 264.7, and primary human macrophage cells and was quantified using qRT-PCR. Total RNA was isolated from $5 \times 10^5$ THP1 cells per well of a 12-well plate, which were pre-incubated with inhibitors for 1 h. Human primary macrophages were used at $3 \times 10^5$ cells per well of a 12-well plate. Cells were transfected using 100 µl of Opti-MEM transfection solution comprising 2 µg of 100-bp dsDNA complexed with 2 µl of Lipofectamine 2000. To assess the specificity of compounds for inhibiting the dsDNA sensing, 0.5 µg ml$^{-1}$ hpRNA Lipofectamine 2000 complex, 10 µg ml$^{-1}$ cGAMP:Lipofectamine 2000 complex, 2 µg ml$^{-1}$ LPS, or 1000 U ml$^{-1}$ recombinant human interferon-β (IFN-β) (PBL assay science) was used. Cells were harvested 4 (**G108, G140**, and **G150**) or 24 h (**J022**) post-transfection, and RNA was extracted using 500 µl of Trizol (Ambion). A total of 800 ng of total RNA was reverse-transcribed for cDNA synthesis in 20 µl final reaction volume using oligo(dT)$_{20}$ primer at 2.5 µM and 10 U µl$^{-1}$ Superscript III (Thermo-Fisher) for 50 min at 50 °C. Primer sequences are shown in Supplementary Table 5. Quantitative PCR was performed on a Mx3000P qPCR System (Agilent Technologies) using 1/20th volume of reverse transcription material as an input for each qPCR reaction. Expression levels of *IFNB1, CXCL10, ISG15, TNF*, and *TUBA1B* mRNAs were measured in technical triplicate for each sample. Threshold cycle ($C_T$) values obtained for the indicated mRNAs were normalized to *TUBA1B* $C_T$ values and used to calculate $\Delta C_T$. Relative mRNA expression levels were calculated using the $\Delta\Delta C_T$ method ($2^{\Delta\Delta C_T}$). The extent of cGAS inhibition was determined by normalizing the mRNA expression level for each sample relative to DMSO only control. qRT-PCR-based *IFNB1* mRNA expression analysis in mouse RAW264.7 cells were performed as described above, except that $2.5 \times 10^5$ cells were plated per well in 12 well plate for 24 h before inhibitor pre-incubation and stimulated with dsDNA and controls for 4 h post transfection.

**Synthesis and characterization of J014, G108, and G140**. Illustrations of reaction schemes for the syntheses of **J014**, **G108**, and **G140** are shown in Supplementary Figures 12 and 13. Detailed synthesis procedures for the compounds and their characterizations are described in Supplementary Methods. **G150** was synthesized using HTOS and, therefore, it was characterized by mass spectrometry and validated from co-crystal structure with h-cGAS.

**Statistical analysis**. All numerical data are expressed as mean ± S.D. of technical replicates of two, three, five or six. For all cellular specificity assays, an unpaired two-tailed Student's *t*-test was used to compare the differences between two groups and one-way ANOVA followed by Tukey's multiple comparison tests was used to compare differences in assays comprising more than two groups. All statistical analyses were performed using GraphPad Prism (7.01).

## Data availability

The pdb files for the solved crystal structures of h-cGAS + **G108/G150**/cGAMP have been deposited in the RCSB Protein Data Bank and can be accessed with the following codes: 6MJU [https://doi.org/10.2210/pdb6MJU/pdb], 6MJW [https://doi.org/10.2210/pdb6MJW/pdb], 6MJX [https://doi.org/10.2210/pdb6KJX/pdb]. The source data underlying the screening results in Fig. 2b have been deposited into NCBI screening library database with the following accession code: 1259398. The source data underlying Figs. 2a, 2c, 8b and c are provided as a Source Data file. All other relevant data that support the findings of this study are available from the corresponding authors upon request.

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

## Acknowledgements

We would like to thank Dr. Paul Bieniasz (Rockefeller University) for providing human blood differentiated macrophages. We thank the synchrotron beam line staff at the Argonne National Laboratory and Brookhaven National Laboratory for their assistance. We thank the High Throughput and Spectroscopy Resource Center at the Rockefeller University for providing equipment infrastructure for compounds screening. The authors gratefully acknowledge the support to the project (not to the non-TDI laboratories) generously provided by the Tri-Institutional Therapeutics Discovery Institute (TDI), a 501(c)(3) organization. TDI receives financial support from Takeda Pharmaceutical Company, TDI's parent institutes (Memorial Sloan Kettering Cancer Center, The Rockefeller University and Weill Cornell Medicine) and from a generous contribution from Mr. Lewis Sanders and other philanthropic sources. Finally, we would like to thank members of Tuschl laboratory for their support, and critical review of the manuscript. This work was supported, in part, by the following agencies: Rockefeller University Robertson Therapeutic Development Funds (J.F.G., T.T., and L.L.), William H. Goodwin and Alice Goodwin from the Commonwealth Foundation for Research and from the Center for Experimental Therapeutics of the Memorial Sloan-Kettering Cancer Center (D.J.P., W.X., and V.K.), and the Leona M. and Harry B. Helmsley Charitable Trust for the purchase of the RF-MS instrument (J.F.G.).

## Author contributions

Biochemical and cell-based assay development for h-cGAS were developed by L.L., C.A., T.G., J.S., T.T. and J.F.G. Compound library screening was performed by C.A., L.L. and J.F.G. Design, curation, and annotation of compound libraries, and screening strategy was performed by J.F.G. Recombinant h- and m-cGAS protein preparation for high-throughput screening and x-ray crystallography was performed by P.G., W.X. and D.J.P. Structure-assisted molecular modeling was performed by V.K., Ma.M., A.J.J., D.J.P. and D.T. The off-target assay using adenylyl cyclase was performed by L.R.E. Chemical syntheses of derivatives of **J001** and **G001** were performed by Mi.M., Y.A., S.H., J.A., T.I., R.O., D.T. and T.Ka. Strategic planning and interpretation of results were conducted at regular meetings coordinated by TDI including contributions of A.S., T.Ku. and P.M. All co-authors contributed to writing and editing the manuscript.

## Additional information

**Competing interests:** T.T., L.L., D.J.P., J.F.G., D.T., T.Ka., Mi.M., Y.A., R.O., S.H., J.A., T.I., Ma.M. and T.Ku. have filed a patent application encompassing aspects of this work (PCT/US2019/016673). The remaining authors declare no competing interests.

