## [Peer Review File · Nature Communications]

Reviewers' comments:

Reviewer #1 (Remarks to the Author):

In this work the authors identify a cGAS inhibitor, and use structural approaches and cellular assays to demonstrate the mode of action and the efficiency/specificity at the cellular level. Given the suggested role of cGAS-STING in human inflammatory diseases, there is a strong interest in the development of a small molecule inhibitor of cGAS. The data presented are generally convincing and the data are fully supported by the data. However, since a cGAS inhibitor has already been reported (Nat Commun. 2017 Sep 29;8(1):750), the data presented in the manuscript lacks depth and demonstration of therapeutical potential, most notably demonstrating of activity in a relevant in vivo model.

MAJOR POINTS

1. The effect of the compound on cGAS activity in vivo should be demonstrated.
2. Recent papers have suggested that cGAS is activated more potently by longer DNA (Nature. 2017 Sep 21;549(7672):394-398; EMBO Rep. 2017 Oct;18(10):1707). Therefore, cGAS activation by longer DNA species may be more difficult to inhibit than shorter DNA species, as used in this study. Therefore, the authors should text whether their compound inhibit cGAS activation by long DNA species and microorganisms.
3. Fig 6g, k, l: The inhibition obtained by G108 is not impressive, and quite high concentrations are needed to achieve more than 90% inhibition of IFN (micromolar range). This weakens the impact of the finding.
4. Fig 6g, k, l: The authors should also measure expression of ISGs. A 60-70% inhibition of IFN might well lead to nearly normal ISG expression, since the IFN levels could be saturated.
5. cGAS-STING activates not only IFN expression, but also NF-kB, autophagy and different types of cell death. It should be evaluated how the inhibitor impact on these pathways.
6. The specificity of the inhibitor is not thoroughly explored. As a minimum it should be examined whether the related OAS family of proteins and other nucleotidyl transferases are targeted by the inhibitor.

Reviewer #2 (Remarks to the Author):

The manuscript focuses on screening and optimizing small molecule inhibitor scaffolds for developing potent inhibitors for cyclic GMP-AMP synthase. A total of four co-crystal structures were obtained to aid the inhibitor development and to provide the molecular details of the h-cGAS and cGAMP interaction. All four structures seemed to be well refined with acceptable statistics.

My overall impression is that the presentation of these structures and their analysis are somewhat sparse, confusing and lack important details. The manuscript also lacks a detailed description on structure- affinity relationship for the lead and final compounds and provide little structural explanation for the improved affinity along the inhibitor optimization (missing the crystal structure with the best compound, G140?). Additionally, structure figures are of low quality with little information on electrostatic environments for the binding pocket, no atomic distances between the receptor (not a single distance is mentioned in the main text and figures!) and the bound small molecule and their fitting in 3D.

1) Figure 4. SRY-fused h-cGAS bound with a dsDNA structure. Fig. 4a, the residue numbers for the SRY-h-cGAS are not spaced correctly and confusing (89160?). Fig. 4b shows a dotted line connecting two domains, but no description of what the dotted line means (no model was built in because e- density is missing?). Need to state what residues are visible and which amino acids are missing in the model. Fig.4d shows structural changes upon ds-DNA binding. While the authors stated that "a minimal conformational change" in the Apo-h-cGAS section, this notion would be

better supported by providing a RMSD value between two structures in the text or the figure legend. Would strengthen the authors' claim better with a RMSD provided for Fig. 4e as well.

2) Figure 5. Shows an inconsistent figure arrangement. 5b needs to be placed next to 5a, not at the below. 5c shows a 2Fo-Fc density without mentioning its sigma level. 5d shows a poor density for the hydroxyl-ethanone, but authors provide no explanation for the possible reason(s). Likely because this moiety provides no direct contact with the pocket? The title for 5e is somewhat confusing. What is "distribution of h-cGAS residues"? It looks more like contact residues for binding G108. The left panel of 5e seems to show a rotated view, but nothing was mentioned in the legend. This figure shows not a single distance, which seems very unusual. For example, the main text mentioned that H437, F488 and L490 are key hydrophobic contacts for the compound without any atomic distance info. 5f would be better represented with an electrostatic surface (APBS). Yellow font is invisible for the compound. Please use black font.

3) X-ray statistics tables. No point separating into two tables and should be combined. Please italicize the letters in the group. No mentioning of the number of the reflections set aside for calculating R_{free}. R-merge seems too high ($> .10$) except for the RU.521 bound structure. The R-merge value for hcGAS-cGAMP seems too low for a 2.44 Å structure and it seems the decimal point is placed wrong. The $I/\sigma I$ values are too low for some structures (0.82 for SRY-h-cGAS-dsDNA and 0.88 for hcGAS-G108) and that means the signal is weaker than the error. Usually an $I/\sigma I$ of 2 is used to decide on the resolution limit. An alternative for cutoff determination is to use CC1/2 values which can better assess data quality and resolution limit (<https://www.ncbi.nlm.nih.gov/pmc/articles/PMC4684713/>). Then the authors should include the CC1/2 values in the statistics table.

4) The validation reports include too many close contacts/bond or Ramachandran outliers except for SRY-h-cGAS-dsDNA complex and need to be reworked (as high as 22 %?). Suggest 3% or lower values?

Reviewer #1 (Remarks to the Author):

In this work the authors identify a cGAS inhibitor, and use structural approaches and cellular assays to demonstrate the mode of action and the efficiency/specificity at the cellular level. Given the suggested role of cGAS-STING in human inflammatory diseases, there is a strong interest in the development of a small molecule inhibitor of cGAS. The data presented are generally convincing and the data are fully supported by the data. However, since a cGAS inhibitor has already been reported (Nat Commun. 2017 Sep 29;8(1):750), the data presented in the manuscript lacks depth and demonstration of therapeutical potential, most notably demonstrating of activity in a relevant in vivo model.

Overall response:

We thank reviewer #1 acknowledging that “there is a strong interest in the development of small molecule cGAS inhibitors”. However, we need to clarify, that the previously reported cGAS inhibitor cited by reviewer 1 (and our previous study) was mouse-specific and shows no cell-based activity against human cGAS and only weak biochemical inhibitory activity. To advance human drug development we focused on obtaining a human-specific compound, but have not been able to develop a potent human and mouse cross-species active inhibitor, and describe here the progress towards a potent human-specific drug. Thus, the best compounds we present in this manuscript preclude testing in established mouse models of interferonopathy or oncogenesis. We included edits to the manuscript to emphasize the differences between mouse versus human and included additional data to show the superiority of our new compound in human cell-based models over our previously reported mouse-specific cGAS inhibitor (Supplementary fig. 10).

MAJOR POINTS**Reviewer #1****Major point 1**

The effect of the compound on cGAS activity in vivo should be demonstrated.

Given the reasons above, we are unable to perform an in vivo study in human yet, but advanced our drug development by showing potent activity against cGAS in human myeloid cells. Studies in human require ADME-toxicology studies and may also require further improvement in potency of the inhibitors. In this manuscript, we document for the first time a class of small molecule inhibitors that specifically works against “human” cGAS enzyme and has activity in “human” myeloid cells.

Major point 2

Recent papers have suggested that cGAS is activated more potently by longer DNA (Nature. 2017 Sep 21;549(7672):394-398; EMBO Rep. 2017 Oct;18(10):1707). Therefore, cGAS activation by longer DNA species may be more difficult to inhibit than shorter DNA species, as used in this study. Therefore, the authors should text whether their compound inhibit cGAS activation by long DNA species and microorganisms.

The dsDNA length dependence on activity of cGAS is noticed only at low dsDNA concentration (0.167 µg/ml) in THP1 cells, a model cellular system for the study of cGAS-STING pathway that we also used (EMBO Rep. 2017 Oct;18(10):1707). At higher dsDNA concentration (1.67 µg/ml), no difference in type I IFN signal strength was observed across the range of DNA sizes from 88 to 4,003 bp. More importantly, the maximum type I IFN signal strength exerted by 4,003 bp DNA at low dsDNA concentration is similar or less to the signal for 88 bp dsDNA at 1.67 µg/ml indicating that the saturation in cGAS activity is observed if 88 bp dsDNA was used at 1.67 µg/ml concentration. This length dependence is also recapitulated in another cellular system (BLaER1) at low dsDNA concentration, however, at higher dsDNA concentration, saturation in cGAS activity is observed for dsDNA 75 bp and above (Nature. 2017 Sep 21;549(7672):394-398).

We used 100 bp dsDNA as applied in our biochemical assay for human cGAS in cellular assays. The dsDNA was used at 2 µg/ml (for qRT-PCR) or 0.5 µg/ml (for luciferase reporter assay) in our cellular assays, at concentrations ensuring maximum cGAS activity by others as well as in our assays as inferred through dsDNA dose response curve.

To address reviewer's comment, we have now included the results for representative cGAS inhibitor **G140** using dsDNA substrates of different lengths and included the results in our revised manuscript (Supplementary fig. 9b).

Major point 3

Fig 6g, k, l: The inhibition obtained by G108 is not impressive, and quite high concentrations are needed to achieve more than 90% inhibition of IFN (micromolar range). This weakens the impact of the finding.

Although the identified inhibitors show nanomolar IC₅₀ values in biochemical assays, the cellular IC₅₀ values are in micromolar concentration, a frequently encountered anomaly during drug development that we believe (and stated in the text) reflects membrane permeability issues of the inhibitors and/or subcellular compartmentalization of cGAS. Nevertheless and more importantly, the inhibitory activities observed are cGAS-specific with minimal off-target effects and cellular toxicity. In order to further convince the reviewer about the relevance of chemotype G class of inhibitors as true cGAS inhibitors, we have included biochemical, structural, and cellular data for another analogue (**G150**) with similar potency as **G140** in our revised manuscript that we identified lately during the course of further investigation into lead optimization.

Major point 4

Fig 6g, k, l: The authors should also measure expression of ISGs. A 60-70% inhibition of IFN might well lead to nearly normal ISG expression, since the IFN levels could be saturated.

Inhibition of ISGs by cGAS inhibitors have previously been assessed in our manuscript using THP1-Dual cells that features a Lucia luciferase reporter gene under the control of an ISG54 minimal promoter in conjunction with five interferon-stimulated response elements. We have addressed reviewer #1's concern with regard to ISGs in our revised manuscript by including data for dose-response analysis of inhibition of another ISG (CXCL10) by cGAS

inhibitors in THP1 cells and primary human macrophages using qRT-PCR based assay (Fig. 6b and 6d).

Major point 5

cGAS-STING activates not only IFN expression, but also NF- κ B, autophagy and different types of cell death. It should be evaluated how the inhibitor impact on these pathways.

In addition to Lucia reporter gene, THP1-Dual cells also express a secreted embryonic alkaline phosphatase (SEAP) reporter gene driven by an IFN- β minimal promoter fused to five copies of the NF- κ B consensus transcriptional response element and three copies of the c-Rel binding site, thereby allowing the study of the NF- κ B pathway, by monitoring the activity of SEAP. We evaluated the impact of inhibitor on NF- κ B pathway using this assay and included the results in Supplementary fig. 9a of our revised manuscript.

Although of consideration for future studies, we are unable to currently measure the impact of inhibitors on other cGAS-mediated pathways such as in autophagy and different types of cell death. As our medicinal chemistry efforts to optimize drug-like properties are ongoing and toxicity and uptake are optimized, we will also broaden our specificity assays. We also believe that progress reported in this manuscript including compounds with cell-based and pathway specific inhibition in human cells in this study, it warrants publication.

Major point 6

The specificity of the inhibitor is not thoroughly explored.

As per reviewer #1's suggestion, our revised manuscript shows results for additional potential off-target effects assessment using other ligands in THP1 cells (Fig. 7c), and primary human macrophages (Fig. 7d-7i).

As a minimum it should be examined whether the related OAS family of proteins and other nucleotidyl transferases are targeted by the inhibitor.

We have now examined possible off-target effect of cGAS inhibitors in the related OAS family of proteins in cellular assay using THP1 cells (Fig. 8a) and primary human macrophages (Fig. 8b). Finally, using an established biochemical assay, we also examined any possible off-target effect of cGAS inhibitors on a representative nucleotidyl transferase enzyme (soluble adenylyl cyclase) and included the results in our revised manuscript as well (Fig. 8c). Our best compounds remain cGAS pathway specific.

Reviewer #2 (Remarks to the Author):

The manuscript focuses on screening and optimizing small molecule inhibitor scaffolds for developing potent inhibitors for cyclic GMP-AMP synthase. A total of four co-crystal structures were obtained to aid the inhibitor development and to provide the molecular details of the h-cGAS and cGAMP interaction. All four structures seemed to be well refined with acceptable statistics.

My overall impression is that the presentation of these structures and their analysis are somewhat sparse, confusing and lack important details. The manuscript also lacks a detailed description on structure-affinity relationship for the lead and final compounds and provide little structural explanation for the improved affinity along the inhibitor optimization (missing the crystal structure with the best compound, G140?).

Overall Response:

We have revised our manuscript by addressing reviewer #2's critiques listed above and itemized below. For clarity, we removed the structure of SRY-fused h-cGAS bound with dsDNA from the revised version (see explanation below). We have also removed the structure of RU.521 bound to h-cGAS^{CD} from the revised version. Instead we included a new structure of the potent inhibitor **G150** (IC₅₀ = 0.010 μM for humans) bound to apo-cGAS^{CD}. We also attempted to solve the structure of potent inhibitor **G140** (IC₅₀ = 0.014 μM for humans) bound to apo-cGAS^{CD} but without success.

We have added further details such as electrostatic representations of the binding pocket and distances between ligand and binding pocket residues. We outline below in a separate section our approach to structure-affinity relationships for the lead and final compounds thereby attempting to provide a structural explanation for the improved affinity along the inhibitor optimization pathway.

Additionally, structure figures are of low quality with little information on electrostatic environments for the binding pocket, no atomic distances between the receptor (not a single distance is mentioned in the main text and figures!) and the bound small molecule and their fitting in 3D.

Please see revised Figs 4c and 5c that provide the electrostatic surface potentials for the binding pockets containing compounds **G108** (Fig. 4c) and **G150** (Fig. 5c). We also list the key atomic distances between residues lining the binding pocket and bound compounds **G108** (Fig. 4e) and **G150** (Fig. 5e).

Reviewer #2

Major point 1

Figure 4. SRY-fused h-cGAS bound with a dsDNA structure. Fig. 4a, the residue numbers for the SRY-h-cGAS are not spaced correctly and confusing (89160?). Fig. 4b shows a dotted line connecting two domains, but no description of what the dotted line means (no model was built in because e- density is missing?). Need to state what residues are visible and which amino acids are missing in the model. Fig.4d shows structural changes upon ds-DNA binding. While the authors stated that "a minimal conformational change" in the Apo-h-cGAS section, this notion would be better supported by providing a RMSD value between two structures in the text or the figure legend. Would strengthen the authors' claim better with a RMSD provided for Fig. 4e as well.

Very recently, a *Cell* paper reported the 2.3 Å resolution structure of a dual mutant h-cGAS-DNA complex¹ as well as some structures of apo h-cGAS bound with small compounds². Thus, the potential of apo h-cGAS to bind small compounds has been validated in the literature and we removed our structure of the SRY-fused h-cGAS-DNA complex from the revised version.

Major point 2

Figure 5. Shows an inconsistent figure arrangement. 5b needs to be placed next to 5a, not at the below.

We have changed the Figure arrangement as requested in the revised version.

5c shows a 2Fo-Fc density without mentioning its sigma level.

We now state that the 2Fo-Fc density is contoured at 1.2 σ in the revised version.

5d shows a poor density for the hydroxyl-ethanone, but authors provide no explanation for the possible reason(s). Likely because this moiety provides no direct contact with the pocket?

The electron density for hydroxyl-ethanone is not observed because it most likely adopts a flexible conformation potentially without direct intermolecular contacts within the binding pocket. This statement is included in the revised version.

The title for 5e is somewhat confusing. What is "distribution of h-cGAS residues"? It looks more like contact residues for binding G108.

The Figure caption has been changed to 'Intermolecular contacts and key distances between **G108** and amino acids lining the binding pocket of h-cGAS^{CD}.

The left panel of 5e seems to show a rotated view, but nothing was mentioned in the legend.

We now point out the extent of rotation in the Figures.

This figure shows not a single distance, which seems very unusual. For example, the main text mentioned that H437, F488 and L490 are key hydrophobic contacts for the compound without any atomic distance info.

We have listed the key atomic distances between the binding pocket residues and bound small molecule for compounds **G108** (Fig. 4e) and **G150** (Fig. 5e).

5f would be better represented with an electrostatic surface (APBS).

We have provided the electrostatic surface potentials for the binding pocket of compounds **G108** (Fig. 4c) and **G150** (Fig. 5c).

Yellow font is invisible for the compound. Please use black font.

We have changed the yellow font to a black font in the revised version.

Major point 3

X-ray statics tables. No point separating into two tables and should be combined. Please italicize the letters in the group.

We have combined the Tables into a single Table and italicized the letters in the group as requested in the revised version.

No mentioning of the number of the reflections set aside for calculating Rfree.

We have added the number of reflections for calculating Rfree in the updated statistics Table in the revised version.

R-merge seems too high ($> .10$) except for the RU.521 bound structure. The R-merge value for hcGAS-cGAMP seems too low for a 2.44 Å structure and it seems the decimal point is placed wrong. The $I/\sigma I$ values are too low for some structures (0.82 for SRY-h-cGAS-dsDNA and 0.88 for hcGAS-G108) and that means the signal is weaker than the error. Usually an $I/\sigma I$ of 2 is used to decide on the resolution limit. An alternative for cutoff determination is to use CC1/2 values which can better assess data quality and resolution limit (<https://www.ncbi.nlm.nih.gov/pmc/articles/PMC4684713/>). Then the authors should include the CC1/2 values in the statistics table.

The x-ray data sets collected at the APS synchrotron were auto-processed by the RAPD online server. Taking into account the reviewer's critique, we have reprocessed all the data sets by HKL2000 using both $I/\sigma I$ and CC21/2 values as the resolution cutoff determinations. Also, we collected a new 2.45 Å resolution x-ray data of compound **G108** bound to h-cGAS (revised Table 5). The $I/\sigma I$ values are 1.5 for h-cGAS-G108, 2.4 for h-cGAS-G150 and 4.5 for h-cGAS-cGAMP. The CC1/2 values are 0.582 for h-cGAS-G108, 0.737 for h-cGAS-G150 and 0.949 for h-cGAS-GAMP.

Major point 4

The validation reports include too many close contacts/bond or Ramachandran outliers except for SRY-h-cGAS-dsDNA complex and need to be reworked (as high as 22 %?). Suggest 3% or lower values?

We included new validation reports for all three structures of the complexes. The clash score and Ramachandran outliers for h-cGAS-G108 are 5.0 and 0%, for h-cGAS-G150 are 4.0 and 0.3% and for h-cGAS-cGAMP are 4.0 and 0%. All Ramachandran outliers now exhibit values below 1%.

Provide a structural explanation for the improved affinity along the inhibitor optimization pathway.

We have responded to this query as outlined below.

A structural explanation for the improved affinity in humans along the inhibitor optimization pathway is difficult to interpret since we have structures of only **G108** and **G150** bound to apo-h-cGAS^{CD}. Our starting point **G001** exhibited a human IC₅₀ = 2.08 μM (Fig. 3d) and improved to 0.35 μM for **G015** (Supplementary Fig. 4b), where the chlorine was moved from position 3 to 2. This 6-fold improvement implies that a chlorine atom at position 2 is beneficial and may reflect hydrogen bond formation involving this electron withdrawing group at position 2 but not at position 3. Replacing the OCH₃ group by OH to form a hydroxyl-ethanone side chain as in **G022** resulted in a further 3.5-fold improvement to IC₅₀ = 0.106 μM (Supplementary Fig. 5). This could imply that the OH group of the hydroxyl-ethanone side chain forms a hydrogen bond with the protein, which is disrupted on O-methylation. The human IC₅₀ = 0.028 μM for **G108** (Fig. 3d) reflects a 4-fold improvement upon addition of a pyrazole ring at position 4. There is space to accommodate a ring system projecting from position 4 allowing for a combination of hydrophobic (with adjacent Leu490 side chains), stacking (with nearby aromatic residues) and hydrogen bonding interactions (pyrazole ring N/NH with a nearby Asn482 side chain). A further 2-fold improvement was observed for **G140** (human IC₅₀ = 0.014 μM; Fig. 3d) and an approx. 3-fold improvement was observed for **G150** (human IC₅₀ = 0.010 μM), suggesting that a combination of additional hydrophobic and hydrogen-bonding interactions accounts for this improvement.

We have added the above paragraph to the Discussion section.

References:

1. Zhou, W. *et al.* Structure of the Human cGAS-DNA Complex Reveals Enhanced Control of Immune Surveillance. *Cell* **174**, 300–311.e11 (2018).
2. Hall, J. *et al.* Discovery of PF-06928215 as a high affinity inhibitor of cGAS enabled by a novel fluorescence polarization assay. *PLoS ONE* **12**, e0184843 (2017).

Reviewers' comments:

Reviewer #1 (Remarks to the Author):

I think the authors have addressed my critics in a satisfactory manner, and I am now convinced that the conclusions are fully supported by the data.

Reviewer #2 (Remarks to the Author):

I appreciate the attempts made by authors to address the major concerns required for improving the quality of the manuscript. I personally feel that the manuscript still requires significant improvements.

The major concerns regarding the revised manuscript are as follows:

1. The X-ray statistics in the revised manuscript have several discrepancies with the values in the PDB validation reports. The authors need to update the X-ray statistics table in the revised manuscript and cross-check the values as per the PDB validation report. Please italicize the letters and not the number in the space group. Please incorporate the R_{sym} and R merge values while PDB coordinate submission. Also, mention the parenthesis in the X-ray statistics table refer to the highest resolution shell.
2. The authors did not mention the change in the numbering of the figures (5 to 4, in the revised manuscript).
3. Figure 5e: The authors state that they "point out the extent of rotation in the Figures". However, it is still not clear in the revised manuscript.
4. Figure 5c: Although, the residues lining the compound-binding pocket are hydrophobic, why the electrostatic potential is positive in the binding pocket. Please clarify and explain/modify it in the manuscript.
5. The structural explanation for the improved affinity along the inhibitor optimization pathway is not convincing and may need additional data to prove their hypothesis.

Reviewer #2 (Remarks to the Author):

I appreciate the attempts made by authors to address the major concerns required for improving the quality of the manuscript. I personally feel that the manuscript still requires significant improvements.

We have taken the concerns of reviewer 2 to heart and substantially revised the overall manuscript, removing redundancy and speculative comments. We have also updated the introduction referring to recent papers and minimized discussion and potential over-interpretation of molecular interactions observed in our 2 crystal structures.

The major concerns regarding the revised manuscript are as follows:

Major point 1

The X-ray statistics in the revised manuscript have several discrepancies with the values in the PDB validation reports. The authors need to update the X-ray statistics table in the revised manuscript and cross-check the values as per the PDB validation report. Please italicize the letters and not the number in the space group. Please incorporate the R_{sym} and R merge values while PDB coordinate submission. Also, mention the parenthesis in the X-ray statistics table refer to the highest resolution shell.

We have finished the updated PDB coordinates submission incorporating the R_{sym} and Rmerge values and got the accession codes: 6MJU for cGAS-G108, 6MJW for cGAS-G150 and 6MJX for cGAS-cGAMP. We have updated the X-ray statistics (Supplementary Table 6) and cross-checked the values with the PDB validation report in the revised version. We have also addressed the remaining queries raised by this reviewer in the revised version.

Major point 2

The authors did not mention the change in the numbering of the figures (5 to 4, in the revised manuscript).

The figure numbering system is corrected in the revised version.

Major point 3

Figure 5e: The authors state that they “point out the extent of rotation in the Figures”. However, it is still not clear in the revised manuscript.

The extent of rotation is now shown in Figures 4c (Figure 4d in the revised manuscript) and 5c (Figure 5d in the revised manuscript).

In addition, we have redrawn Figures 4e and 5e so that they include the same amino acids in both panels.

Major point 4

Figure 5c: Although, the residues lining the compound-binding pocket are hydrophobic, why the

electrostatic potential is positive in the binding pocket. Please clarify and explain/modify it in the manuscript.

We have redrawn Figures 4c (Figure 4d in the revised manuscript) and 5c (Figure 5d in the revised manuscript) indicating higher thresholds (± 10 kcal/mol-e) depicting the electrostatic surface representation using the Coulombic Surface tool in Chimera. The hydrophobicity of segments lining the binding pocket are more clearly visible in these redrawings.

Major point 5

The structural explanation for the improved affinity along the inhibitor optimization pathway is not convincing and may need additional data to prove their hypothesis.

We now monitored the inhibitor activity of two catalytic pocket mutants h-cGAS (N482H) and h-cGAS (Y248F) where the human residues were replaced by their mouse counterparts (Supplementary Fig 7). Given that N482H showed no enzymatic activity, we focused on the Y248F mutant. We observed a strong reduction in inhibitory activity for this mutant, implying that Y248 is important for inhibitor specificity. Indeed, we observed a hydrogen bond between the N1 position of 2-amino pyridine ring of G150 and hydroxyl group of Y248 in the G150 inhibitor complex (Fig 5e).

Overall, we also minimized any discussion or reference to over-interpret isolated molecular interactions between drug and amino acids or speculation regarding differences of inhibition between mouse and human enzyme.

REVIEWERS' COMMENTS:

Reviewer #2 (Remarks to the Author):

I feel that Authors addressed all of the comments successfully. I very much appreciate the efforts made by authors and their thoroughness in addressing the comments. I congratulate all contributing authors for the wonderful story.